# Dystrophy-associated caveolin-3 mutations reveal that caveolae couple IL6/STAT3 signaling with mechanosensing in human muscle cells

Melissa Dewulf[1], Darius Vasco Köster [2], Bidisha Sinha[3], Christine Viaris de Lesegno[1], Valérie Chambon[4], Anne Bigot[5], Mona Bensalah[5], Elisa Negroni[5], Nicolas Tardif[1], Joanna Podkalicka[1,6,7], Ludger Johannes [4], Pierre Nassoy[8], Gillian Butler-Browne[5], Christophe Lamaze[1,9] & Cedric M. Blouin[1,9]

Caveolin-3 is the major structural protein of caveolae in muscle. Mutations in the *CAV3* gene cause different types of myopathies with altered membrane integrity and repair, expression of muscle proteins, and regulation of signaling pathways. We show here that myotubes from patients bearing the *CAV3* P28L and R26Q mutations present a dramatic decrease of caveolae at the plasma membrane, resulting in abnormal response to mechanical stress. Mutant myotubes are unable to buffer the increase in membrane tension induced by mechanical stress. This results in impaired regulation of the IL6/STAT3 signaling pathway leading to its constitutive hyperactivation and increased expression of muscle genes. These defects are fully reversed by reassembling functional caveolae through expression of caveolin-3. Our study reveals that under mechanical stress the regulation of mechanoprotection by caveolae is directly coupled with the regulation of IL6/STAT3 signaling in muscle cells and that this regulation is absent in Cav3-associated dystrophic patients.

[1] Membrane Dynamics and Mechanics of Intracellular Signaling Laboratory, Institut Curie – Centre de Recherche, PSL Research University, CNRS UMR3666, INSERM U1143, Paris 75248, France. [2] Centre for Mechanochemical Cell Biology and Division of Biomedical Sciences, Warwick Medical School, University of Warwick, Coventry CV4 7AL, UK. [3] Department of Biological Sciences, Indian Institute of Science Education and Research (IISER) Kolkata, Mohanpur 741 246 West Bengal, India. [4] Endocytic Trafficking and Intracellular Delivery Laboratory, Institut Curie – Centre de Recherche, PSL Research University, CNRS UMR3666, INSERM U1143, Paris 75248, France. [5] Association Institut de Myologie, Centre de Recherche en Myologie, Sorbonne Université, INSERM, UMRS974, Paris 75013, France. [6] Laboratoire Physico-Chimie Curie, Institut Curie, PSL Research University, CNRS UMR168, Sorbonne Université, Paris 75231, France. [7] Laboratory of Cytobiochemistry, Faculty of Biotechnology, University of Wrocław, Wrocław 50-383, Poland. [8] LP2N, CNRS UMR 5298, IOA, Institut d'Optique Graduate School, Université de Bordeaux, Talence 33400, France. [9] These authors jointly supervised this work: Christophe Lamaze, Cedric M. Blouin. Correspondence and requests for materials should be addressed to C.L. (email: christophe.lamaze@curie.fr) or to C.M.B. (email: cedric.blouin@curie.fr)

Caveolae are cup-shaped plasma membrane invaginations that were first observed in the 1950s by Palade and Yamada on electron micrographs from vascular and gall bladder tissues[1,2]. Caveolae present a specific protein signature involving two main families of proteins, caveolins (caveolin-1, -2, and -3), and cavins (cavin-1, -2, -3, and -4)[3–9]. Caveolins and cavins are expressed in almost every cell type, except for caveolin-3 (Cav3) and cavin-4, whose expression is restricted to smooth and striated muscle cells[9,10]. Cav3, like Cav1 in non-muscle cells, is necessary for the formation of caveolae at the plasma membrane of muscle cells[11].

Caveolae have long been associated with several important cellular functions including endocytosis, lipid metabolism, and cell signaling, albeit with several persistent controversies[12,13]. More recently, a new function of caveolae was established as mechanosensors that play an essential role in cell mechanoprotection both in vitro and in vivo[14–18]. Mutations or abnormal expression of caveolae components have been associated with lipodystrophy, vascular dysfunction, cancer, and muscle disorders[13,19]. The molecular mechanisms underlying caveolin-associated diseases are still poorly understood.

In this study, we explored the mechanical role of caveolae in human muscle cells and their possible deregulation in caveolinopathies, a family of muscle genetic disorders involving mutations in the CAV3 gene. These diseases affect both cardiac and skeletal muscle tissues, and share common characteristics including mild muscle weakness, high levels of serum creatine kinase, variations in muscle fiber size, and an increased number of central nuclei[20–23]. We focused our investigations on the human CAV3 P28L mutation responsible for hyperCKemia[24], and CAV3 R26Q, which is responsible for ripple muscle disease, hyperCKemia, and limb-girdle muscular dystrophy 1C[25]. Studies with transgenic mice and zebrafish or cells overexpressing the Cav3 mutants have linked the P28L and R26Q CAV3 mutations to deregulations in distinct signaling pathways[22,25,26], defects in membrane repair[27,28], and mechanoprotection of the muscle tissue[16]. Nevertheless, the role of the caveolae mechanoresponse in human myotubes and its possible deregulation in dystrophy-associated Cav3 mutations have not yet been addressed.

We show here that the Cav3 P28L and Cav3 R26Q myotubes are unable to assemble sufficient amounts of functional caveolae at the plasma membrane, leading to a loss of membrane tension buffering and membrane integrity under mechanical stress. The absence of functional caveolae in mutant myotubes uncouples the regulation of IL6/STAT3 signaling with mechanical stress, which results in the constitutive hyperactivation of the IL6/STAT3 signaling pathway and the upregulation of several muscle-related genes. Finally, the expression of WT Cav3 in mutant myotubes is sufficient to restore a functional pool of caveolae and to rescue the coupling of caveolae mechanosensing with IL6/STAT3 signaling. These results establish caveolae as central connecting devices that adapt intracellular signaling to mechanical cues in muscle cells. The loss of this function in Cav3-associated mutations may be responsible for some of the clinical symptoms described in human dystrophic patients.

## Results

### Decrease of caveolae number in Cav3 mutant myotubes.

To address the impact of CAV3 mutations in human muscle disorders, we analyzed wild type (WT), Cav3 P28L, and Cav3 R26Q myotubes derived from immortalized myoblasts, which were isolated from healthy or Cav3 mutant patients and differentiated for 4 days. The state of myotube differentiation was validated by the expression level of the differentiation marker MF20 (myosin heavy chain) in all three cell lines (Supplementary Fig. 1a). We first analyzed the presence and the ultrastructure of caveolae at the plasma membrane of myotubes by electron microscopy. In WT myotubes, we observed numerous invaginated structures corresponding to bona fide caveolae i.e., characteristic 60–100 nm cup-shaped invaginations that were connected to the plasma membrane, or to larger vacuoles of variable size deeper inside the cell known as rosettes, and that could still be connected to the plasma membrane. In contrast, a lot less caveolae could be detected at the plasma membrane of mutant myotubes and very few, if any, large vacuolar structures were observed (Fig. 1a, b). While we could still visualize a few caveolae in mutant myotubes, they were often grouped in the same area and large areas of plasma membrane were completely devoid of caveolae (not shown). Interestingly, we could observe, mainly in mutant myotubes, the presence of aberrant oversized caveolae (Fig. 1a).

This drastic decrease in caveolae number led us to investigate the localization of Cav3, which is required for caveolae assembly at the plasma membrane[11]. Immunoblot analysis showed a reduced expression of mutant Cav3 (P28L: −50%; R26Q: −51%) as compared to WT (Fig. 1c), with a shifted band for the R26Q mutant corresponding to the Cav3 mutant form, as reported previously[25]. Cav3 immunostaining revealed that WT Cav3 was mainly associated with the plasma membrane of myotubes and partially localized in the Golgi complex, defined by GM130 staining (Fig. 1d). In contrast, Cav3 strongly accumulated in the Golgi complex as shown by the colocalization with GM130 in the Cav3 P28L and R26Q myotubes, in agreement with earlier studies[25,26]. This indicates that the strong reduction in the number of caveolae present at the plasma membrane of the Cav3 mutant myotubes is a consequence of the abnormal retention of mutant Cav3 in the Golgi complex.

Since myoblasts express Cav1 but not Cav3, differentiated myotubes might still express Cav1, which could potentially participate in the formation of caveolae independently from Cav3. We therefore analyzed Cav1 expression in myotubes after 4 days of differentiation and found that Cav1 was indeed expressed to the same level in all three cell lines (Supplementary Fig. 1b). Cav1 colocalized perfectly with Cav3 at the plasma membrane and to a lesser extent at the Golgi complex in WT myotubes, whereas it was mainly present in the Golgi complex in Cav3 P28L and R26Q myotubes (Supplementary Fig. 1c, d). Finally, we used giant plasma membrane vesicles (GPMVs) isolated from myotubes to unambiguously determine the localization of Cav1 and Cav3 since GPMVs are composed of plasma membrane and cytosol while excluding other subcellular compartments[29] (see Supplementary Methods). GPMVs biochemical analysis confirmed that Cav3 was not present at the plasma membrane of Cav3 P28L and R26Q myotubes. While Cav1 was also present at the plasma membrane of WT myotubes, it was much less abundant in mutant myotubes (Supplementary Fig. 1e). Altogether, these data indicate that Cav1 is likely to form hetero-oligomers with Cav3, and that the Cav3 P28L and R26Q mutants have a dominant effect on Cav1 localization.

### Tension buffering and mechanoprotection defects in Cav3 mutants.

To know whether the almost total absence of caveolae at the plasma membrane of mutant myotubes could induce defects in cell mechanoprotection, we first determined if the Cav3 P28L and R26Q myotubes could buffer the increase of membrane tension induced by mechanical stress. We thus applied a 45 mOsm hypo-osmotic shock to myotubes aligned by micro-patterning and we measured the apparent membrane tension before and after 5 min of hypo-osmotic shock using membrane nanotube pulling with optical tweezers as described[14]. As expected, hypo-osmotic shock led to myotube swelling in WT and Cav3 mutant cells (Fig. 2b). While mutant myotubes showed

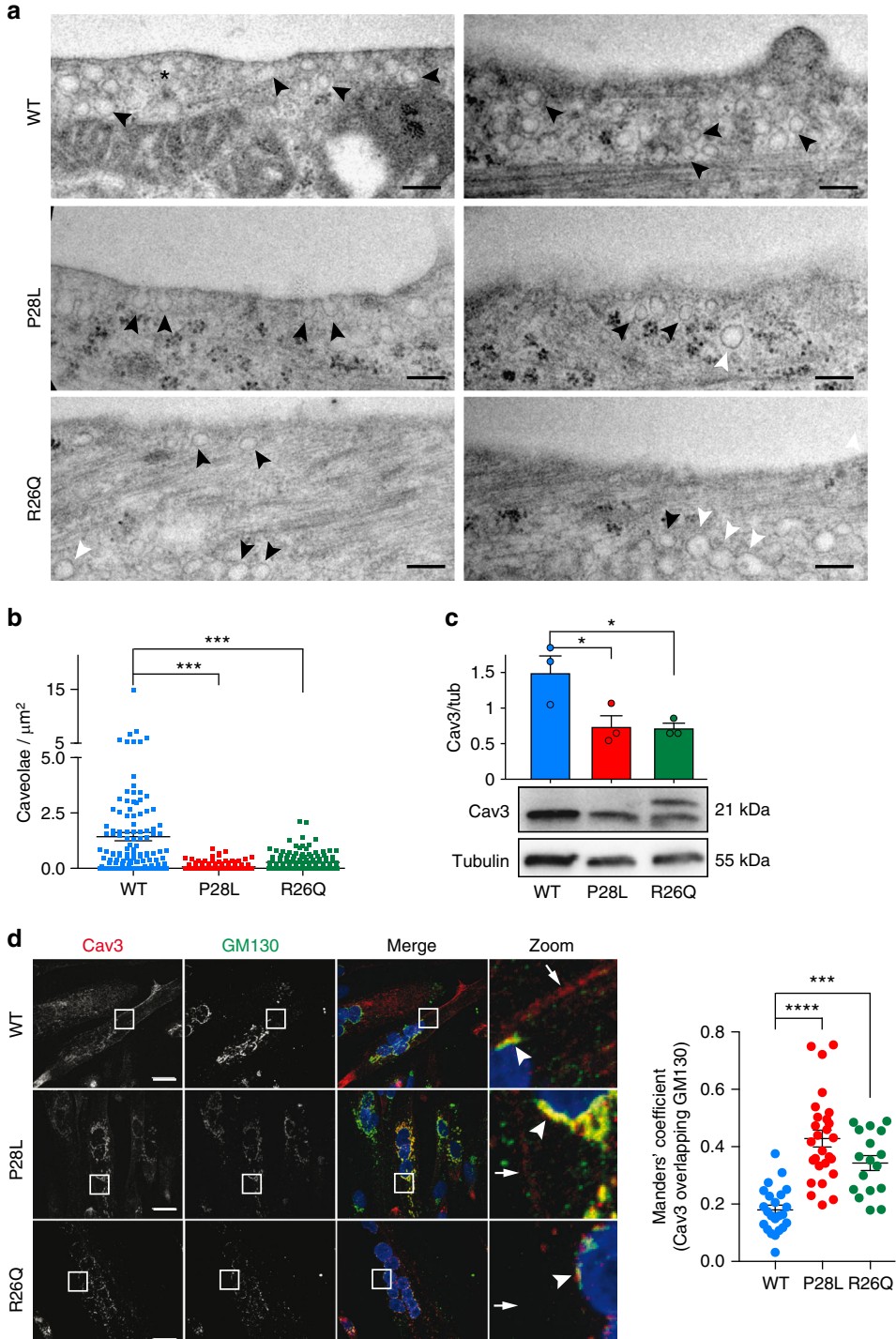

**Fig. 1** Characterization of caveolae and Cav3 expression in WT, Cav3 P28L, and Cav3 R26Q myotubes. **a** Electron micrographs of WT, Cav3 P28L, and Cav3 R26Q myotubes. Caveolae, interconnected caveolae, and aberrant sized caveolae are indicated with black arrowheads, asterisks, and white arrowheads, respectively. **b** Quantification of the number of caveolae/$\mu m^2$ in **a**. **c** Immunoblot analysis (lower panel) and quantification (upper panel) of total levels of Cav3 in WT, Cav3 P28L, and Cav3 R26Q differentiated myotubes. Tubulin serves as a loading control. Quantification of the expression of Cav3 by calculating the ratio between Cav3 and tubulin expression. **d** Immunofluorescent labeling of Cav3 and GM130 in WT, Cav3 P28L, or Cav3 R26Q myotubes analyzed by confocal microscopy. Arrows in inset indicate the plasma membrane and arrowheads indicate the Golgi complex. Cav1 staining is shown in Supplementary Fig. 1c. **a** Scale bar = 200 nm. Representative cells quantified in **b** (number of regions analyzed: WT = 115, P28L = 154, R26Q = 146; total area screened: WT = 1140 $\mu m^2$, P28L = 1187 $\mu m^2$, R26Q = 1216 $\mu m^2$). Reproducibility of experiments: **a** Representative cells. **a**, **c**, **d** Representative data. **b**, **c** Quantification was done on 3 independent experiments. **d** Quantification was done on 3 independent experiments (WT $n = 24$ cells, P28L $n = 27$ cells, R26Q $n = 17$ cells). Mean value ± SEM. **b**, **c** Statistical analysis with a two-tailed unpaired $t$ test. **d** Statistical analysis with a one-way ANOVA *$P < 0.05$; ***$P < 0.001$; ****$P < 0.0001$

no significant changes in membrane tension in resting condition (Fig. 2a), they showed a significant increase of membrane tension (P28L: 63 ± 7%; R26Q: 94 ± 11%) under 45 mOsm hypo-osmotic

shock compared to WT myotubes (38 ± 9%) (Fig. 2b). These results clearly show that the Cav3 P28L and R26Q mutant

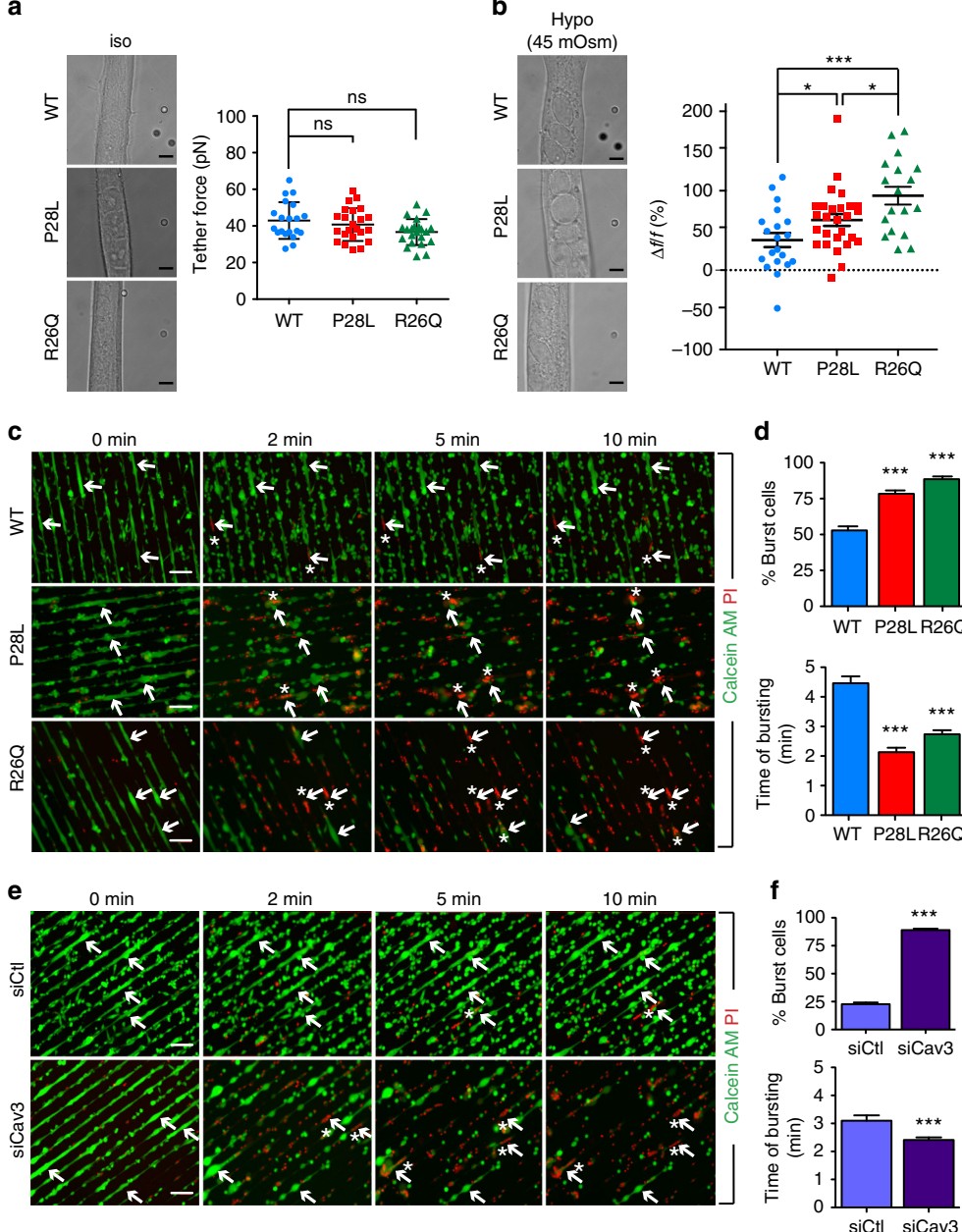

**Fig. 2** Cav3 mutant myotubes present major defects in membrane tension buffering and integrity. **a**, **b** Membrane tension measurement analysis using optical tweezers and nanotube pulling on micropatterned WT, Cav3 P28L, or Cav3 R26Q myotubes. Membrane tethers were pulled in the perpendicular axis of aligned myotubes after micropatterning in resting conditions and 5 min after a 45 mOsm hypo-osmotic shock (**a**, **b**, left panels). Membrane tension was analyzed in resting condition (**a**, right panel) and the difference of membrane tension before and after hypo-osmotic shock was calculated, reflecting the percentage of increase of membrane tension upon mechanical stress (**b**, right panel). **c**, **e** Micropatterned WT, Cav3 P28L, or Cav3 R26Q myotubes (**c**) or WT ctl (siCtl) and Cav3-depleted (siCav3) myotubes (**e**) were loaded with calcein-AM (green). The medium was switched with a 30 mOsm medium supplemented with propidium iodide (PI, red). Representative pictures were taken at the indicated times during hypo-osmotic shock. Arrows correspond to myotubes and asterisks correspond to burst myotubes. **d**, **f** Quantification of the percentage of burst myotubes (upper panel) and mean time of bursting in minutes (lower panel) in **c** and **e**, respectively. **a**, **b** Scale bar = 5 μm. **c**, **e** Scale bar = 120 μm. Reproducibility of experiments: **a** Representative pictures and quantifications from 7 independent experiments (WT $n = 20$ cells, P28L $n = 23$ cells, and R26Q $n = 22$ cells). **b** Representative pictures and quantifications from 7 independent experiments (WT $n = 20$ cells, P28L $n = 27$ cells, and R26Q $n = 18$ cells). **c** Representative data of 3 independent experiments quantified in **d** (% burst cells: WT $n = 310$ cells, P28L $n = 299$ cells, and R26Q $n = 271$ cells; mean time of bursting: WT $n = 165$ cells, P28L $n = 233$ cells, and R26Q cells $n = 240$ cells). **e** Representative data of 3 independent experiments quantified in **f** (% burst cells: siCtl $n = 749$ cells and siCav3 $n = 569$ cells; mean time of bursting: siCtl $n = 171$ cells and siCav3 $n = 506$ cells). **a**, **d**, **f** Mean value ± SD. **b** Mean value ± SEM. **a**, **b** Statistical analyses were done using Kruskal–Wallis test. **d**, **f** Statistical analysis with two-tailed unpaired $t$ test; *$P < 0.05$; ***$P < 0.001$

myotubes have lost the ability to buffer membrane tension variations induced by mechanical stress.

We next tested whether the lack of membrane tension buffering could result in insufficient mechanoprotection and increased membrane fragility in mechanically challenged mutant myotubes. We designed an assay to quantify the percentage of cells that rupture their membrane under mechanical stress. To monitor membrane bursting, micropatterned myotubes were incubated with calcein-AM, a permeant green fluorescent dye that only becomes fluorescent inside the cell, and with the nucleus-specific blue dye DAPI to specifically visualize differentiated myotubes by nucleus staining (Supplementary Fig. 2a). Live imaging was performed on myotubes subjected to a 30 mOsm hypo-osmotic shock for 10 min in the presence of propidium iodide (PI), a non-permeant red fluorescent dye that cannot enter cells with intact plasma membrane. The concomitant decrease of calcein-AM fluorescence and the appearance of PI fluorescence in the nucleus indicate a loss of membrane integrity (Fig. 2c). In comparison to WT myotubes, Cav3 mutant myotubes not only showed a higher percentage of burst cells after a 10 min hypo-osmotic shock (WT: $53 \pm 3\%$; P28L: $78 \pm 2\%$; R26Q: $89 \pm 2\%$) but also a shorter time of resistance to membrane bursting (WT: $4.5 \pm 0.2$ min, P28L: $2.1 \pm 0.1$ min, R26Q: $2.7 \pm 0.2$ min) (Fig. 2d). When we apply a milder hypo-osmotic shock (150 mOsm), for which no increase in membrane tension could be measured, the plasma membrane of all three cell lines remained intact after 10 min of shock (Supplementary Fig. 2b, c). We repeated these experiments in WT myotubes depleted for Cav3 and measured a percentage of burst cells that was similar to mutant myotubes (siCtl: $23 \pm 1\%$, siCav3: $89 \pm 1\%$) (Fig. 2e, f; Supplementary Fig. 2d). Likewise, Cav3-depleted myotubes showed a significantly faster time of bursting as compared to control myotubes (siCtl: $3.1 \pm 0.2$ min, siCav3: $2.4 \pm 0.1$ min) (Fig. 2e, f). Together, our results demonstrate that the Cav3 P28L and R26Q mutant myotubes are unable to provide the mechanoprotection that is required to maintain the integrity of the myotube plasma membrane under mechanical stress and behave similarly to myotubes depleted for Cav3.

**Hyperactivation of IL6/STAT3 signaling in Cav3 mutant myotubes.** Considering the key role of caveolae and caveolin in intracellular signaling[12,13], we next investigated whether the loss of functional caveolae could impact some of the key signaling pathways in the muscle. We focused our analysis on the IL6/STAT3 signaling pathway that has been associated with satellite cell exhaustion and muscle wasting[30–32]. Furthermore, the IL6 signal transducer glycoprotein gp130, which, together with the IL6 receptor subunit, assemble the IL6 receptor, has been localized in caveolae in a myeloma cell line[33], suggesting a potential regulation of the IL6 signaling pathway by caveolae. IL6 binding to the IL6 receptor is classically followed by the activation of receptor-bound JAK1 and JAK2 kinases, which in turn phosphorylate the signal transducer and activator of transcription 3 (STAT3) that is then translocated as a dimer to the nucleus where it activates the transcription of IL6 sensitive genes[34].

We therefore monitored the level of STAT3 activation, i.e., tyrosine (Tyr705) phosphorylation (pSTAT3) in myotubes stimulated for 5 and 15 min with physiological concentrations of IL6 (Fig. 3). At steady state, in the absence of IL6 stimulation, little tyrosine phosphorylation of STAT3, if any, could be detected in WT myotubes. In contrast, we found a substantially higher level of pSTAT3 in Cav3 P28L and R26Q mutant myotubes, even in the absence of IL6 stimulation. While IL6 stimulation led to increased levels of pSTAT3 in WT myotubes, we still observed higher levels of pSTAT3 in Cav3 P28L and R26Q mutant

myotubes for similar times of IL6 stimulation (Fig. 3a, b). We could rule out the higher expression of STAT3 in Cav3 P28L mutant myotubes as responsible for STAT3 hyperactivation since Cav3 R26Q mutant myotubes, which express STAT3 at levels identical to WT myotubes, showed the same hyperactivation of STAT3. To eliminate the possible contribution of undifferentiated myotubes in IL6 signaling, we investigated the nuclear translocation of pSTAT3 by immunofluorescence since differentiated myotubes are characterized by the presence of multiple nuclei (Fig. 3c, d). Again, we detected a significantly higher level of pSTAT3 in the nuclei of mutant myotubes as compared to WT at steady state. After 15 min of IL6 stimulation, mutant myotubes exhibited higher pSTAT3 nuclear translocation, although it was less pronounced in P28L mutants. Altogether, these data reveal that the IL6/STAT3 signaling pathway is constitutively hyperactivated in the Cav3 P28L and R26Q mutant myotubes.

We next investigated whether the regulation of IL6/STAT3 signaling would require the presence of functional caveolae at the plasma membrane and thus the expression of Cav3. We therefore monitored the kinetics of STAT3 activation by IL6 in WT myotubes depleted for Cav3. Immunoblot analysis showed a hyperactivation of the IL6 pathway in Cav3-depleted myotubes with an overall activation of STAT3 (Fig. 3e, f). In contrast to Cav3 P28L and R26Q myotubes, STAT3 was not activated in the absence of IL6 stimulation. This could be due to the compensation of the loss of Cav3 by Cav1 at the plasma membrane since it is unlikely that Cav1 will be retained in the Golgi apparatus in siCav3 WT myotubes. These results indicate that Cav3 is a negative regulator of the IL6/STAT3 pathway in healthy myotubes and that the depletion of Cav3 in WT myotubes reproduces to some extent the phenotype observed in the Cav3 mutants. It suggests that the absence of Cav3 and/or caveolae at the plasma membrane of mutant myotubes is responsible for the constitutive hyperactivation of the IL6/STAT3 signaling pathway.

STAT3 is a key transcription factor controlling the transcription of many downstream genes whose products mediate the pleiotropic effects of STAT3 in physiological and pathological contexts[35]. We therefore examined the consequences of the constitutive hyperactivation of the IL6/STAT3 pathway on gene expression. In the context of muscle disease, we investigated the transcription of muscle-related genes since STAT3 has been suggested to be involved in their regulation. We focused our analysis on the SOCS3, MYH8, ACTC1, and ACTN2 genes that are associated with muscle development and regeneration[32]. SOCS3 serves also as a positive control, as it is transcribed upon STAT3 activation and its gene product SOCS3 is a major actor in the negative regulation of this pathway[34]. Using quantitative PCR, we found an increased transcription of SOCS3, MYH8, ACTC1, and ACTN2 genes (Fig. 3g). These data strongly suggest that the constitutive hyperactivation of STAT3 found in the Cav3 P28L and R26Q mutant myotubes is responsible for the deregulation of several genes involved in muscle pathophysiology.

**IL6/STAT3 mechanosignaling is impaired in Cav3 mutant myotubes.** Although caveolae and caveolins have long been associated with signaling[12,13], the integration of this function with their role in mechanosensing has not yet been reported. We have proposed the hypothesis that the mechano-dependent cycle of caveolae disassembly and reassembly could impact some of the caveolae-dependent signaling pathways[13,36]. We thus analyzed whether the regulation of the IL6/STAT3 pathway by caveolae could depend on mechanical stress. When myotubes were subjected to hypo-osmotic shock prior to IL6 stimulation, we observed a dramatic decrease of STAT3 activation (approx. 80%) in WT myotubes whereas no significant change was observed in Cav3 P28L and R26Q mutant

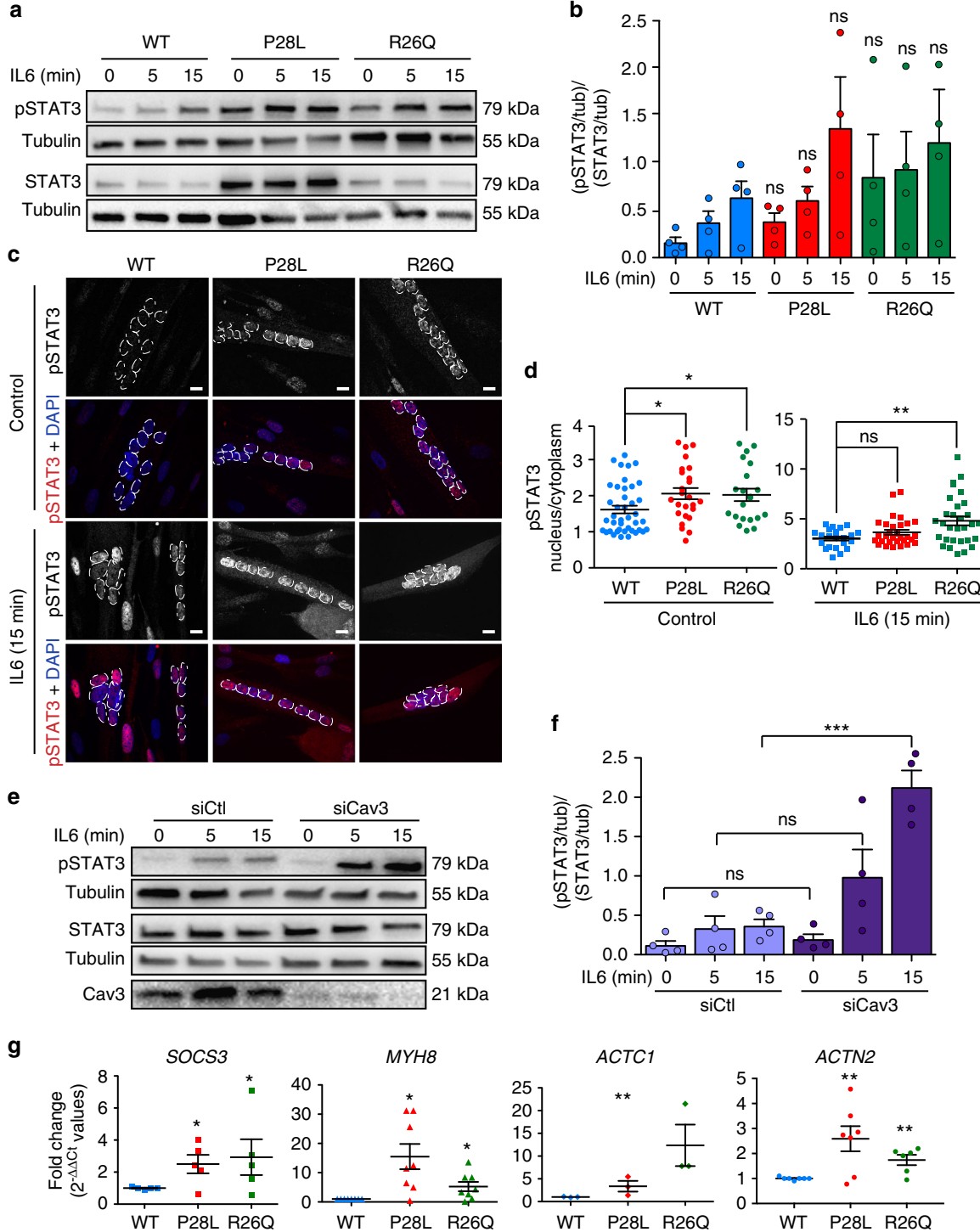

**Fig. 3** Constitutive hyperactivation of IL6/STAT3 signaling in Cav3 mutant myotubes. **a** Immunoblot analysis of pSTAT3 and STAT3 levels in WT, Cav3 P28L, and Cav3 R26Q myotubes stimulated for the indicated times with 10 ng mL$^{-1}$ IL6. Tubulin serves as a loading control. **b** Quantification of STAT3 activation of **a**, corresponding to the ratio pSTAT3 on STAT3 total levels after normalization to tubulin levels. **c** Confocal microscopy of immunofluorescent pSTAT3 in WT, Cav3 P28L, and Cav3 R26Q myotubes stimulated or not for 15 min with 10 ng mL$^{-1}$ IL6. White dashed lines outline nucleus boundaries. **d** Quantification of pSTAT3 nuclear translocation in **c** corresponding to nuclei/cytoplasm mean intensity ratio of pSTAT3. **e** Immunoblot analysis of pSTAT3 levels in WT ctl (siCtl) and Cav3-depleted (siCav3) myotubes stimulated for the indicated times with 10 ng mL$^{-1}$ IL6. **f** Quantification of STAT3 activation in **e**, corresponding to the ratio pSTAT3 on STAT3 total level after normalization with tubulin level. **g** Expression of STAT3 related genes: from left to right *SOCS3*, *MYH8*, *ACTC1*, and *ACTN2* in WT, Cav3 P28L, or Cav3 R26Q myotubes. **c** Scale bar = 10 μm. Reproducibility of experiments: **a**, **c**, **e** Representative data. **b** Quantification was done on 4 independent experiments. **d** Quantification was done on 3 independent experiments (0 min: WT n = 41 cells, P28L n = 25 cells, R26Q n = 21 cells; 15 min: WT n = 22 cells, P28L n = 30 cells, R26Q n = 30 cells). **f** Quantification was done on 4 experiments. **g** Quantification was done on 5 (*SOCS3*), 8 (*MYH8*), 3 (*ACTC1*), and 7 (*ACTN2*) independent experiments. Mean value ± SEM. **b**, **f** Statistical analysis with two-tailed paired *t* test. **d**, **g** Statistical analysis with two-tailed unpaired *t* test *$P < 0.05$; **$P < 0.01$; ***$P < 0.001$; ns non-significant

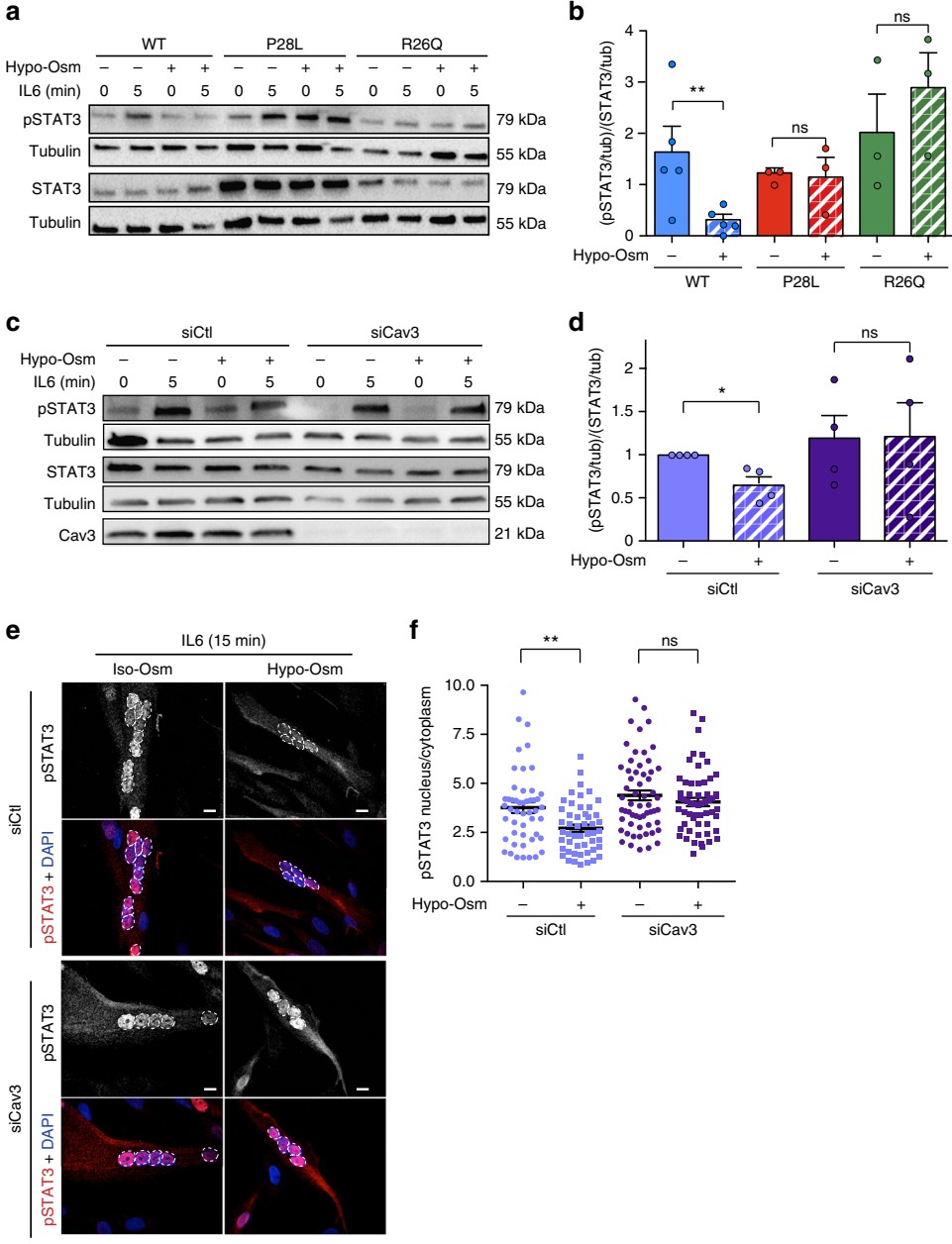

**Fig. 4** IL6/STAT3 mechanosignaling is impaired in Cav3 P28L and R26Q myotubes. **a, c** Immunoblot analysis of pSTAT3 and STAT3 levels in WT, Cav3 P28L, and Cav3 R26Q myotubes (**a**), and WT ctl (siCtl) or Cav3-depleted (siCav3) myotubes (**c**) subjected or not to a 75 mOsm hypo-osmotic shock (Hypo-Osm) for 10 min, followed by stimulation or not with 10 ng mL$^{-1}$ IL6 for 5 min. Tubulin serves as loading control. **b, d** Quantification of STAT3 activation in **a** and **c** respectively, corresponding to the ratio pSTAT3 on STAT3 total level after normalization to tubulin level. **e** Confocal microscopy of immunofluorescent pSTAT3 in WT (siCtl) and Cav3-depleted (siCav3) myotubes subjected or not to a 75 mOsm hypo-osmotic shock (Hypo-Osm) for 5 min, followed by stimulation or not with 10 ng mL$^{-1}$ IL6 for 15 min. White dashed lines outline nucleus boundaries. **f** Quantification of pSTAT3 nuclear translocation in **e** corresponding to the mean intensity of the nuclei/cytoplasm pSTAT3 ratio. **e** Scale bar = 10 μm. Reproducibility of experiments: **a, c, e** Representative data. **b** Quantification was done on 5 and 3 independent experiments for WT and mutants, respectively. **d** Quantification was done on 4 independent experiments. **f** Quantification was done on 3 independent experiments (Iso-Osm: siCtl $n = 49$ cells, siCav3 $n = 50$ cells; Hypo-Osm: siCtl $n = 58$ cells, siCav3 $n = 57$ cells). Mean value ± SEM. **b, d** Statistical analysis with two-tailed paired $t$ test. **d** Statistical analysis with a one-way ANOVA *$P < 0.05$; **$P < 0.01$; ns non-significant

myotubes (Fig. 4a, b). We also tested the effect of mechanical stretching on IL6/STAT3 signaling as this is more relevant to the nature of mechanical stress experienced by skeletal muscles during exercise. When we applied a 10% cyclic stretch at 0.5 Hz for 30 min to WT myotubes followed by IL6 stimulation, we also observed a drastic reduction of STAT3 activation, confirming that the IL6/STAT3 pathway is tightly regulated by mechanical stress in muscle cells (Supplementary Fig. 3).

We next determined whether the mechanical regulation of IL6 signaling required the presence of functional caveolae. We applied a hypo-osmotic shock to WT myotubes depleted of Cav3 and whereas no effect was observed at steady state, we found that STAT3 activation by IL6 was slightly decreased by mechanical stress (approx. 20%) in WT myotubes. The decrease of STAT3 activation was more pronounced when the analysis was restricted to the translocation of STAT3 in the nuclei of fully differentiated

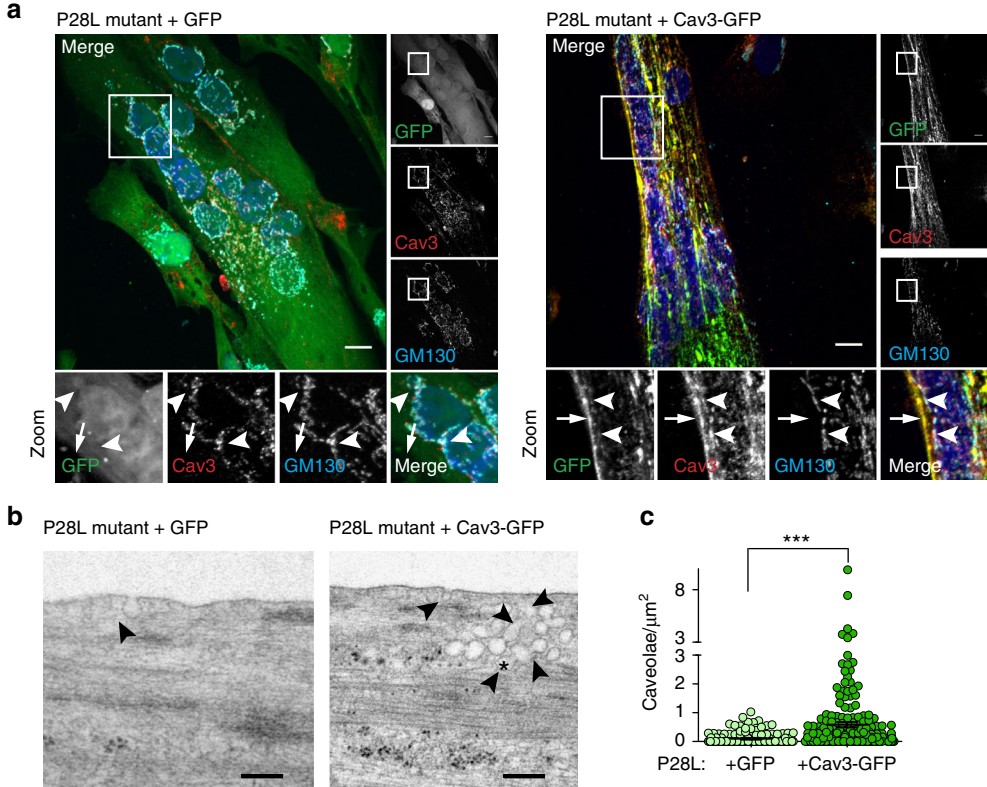

**Fig. 5** Expression of WT Cav3 restores the formation of caveolae at the plasma membrane of Cav3 P28L myotubes. **a** Immunofluorescent labeling of Cav3 and Golgi marker GM130 in Cav3 P28L GFP and P28L Cav3-GFP transduced myotubes analyzed by confocal microscopy. Arrows in the inset indicate the plasma membrane and arrowheads indicate the Golgi complex. **b** Electron micrographs of Cav3 P28L GFP and P28L Cav3-GFP transduced myotubes. Caveolae and interconnected caveolae are indicated with arrowheads and asterisks, respectively. **c** Quantification of the number of caveolae/$\mu m^2$ in **b**. **a** Scale bar = 10 μm. **b** Scale bar = 200 nm. Reproducibility of experiments: **a** Representative pictures of 3 experiments. **b** Representative pictures quantified in **c** (number of analyzed regions: P28L GFP = 169, P28L Cav3-GFP = 182; total screened area: P28L GFP = 1405 μm², P28L Cav3-GFP = 1349 μm²). Mean value ± SEM. Statistical analysis with a two-tailed unpaired *t* test; \*\*\*$P < 0.0001$; ns non-significant

myotubes so as to rule out a possible contamination by myoblasts (Fig. 4e, f). More importantly, no changes were observed in Cav3-depleted myotubes (Fig. 4c–f). The poor adhesion of Cav3 P28L and R26Q mutant myotubes on the stretching membrane did not allow us to confirm these data under cyclic stretching. Nevertheless, these results confirm that the IL6/STAT3 signaling pathway is negatively regulated by mechanical stress in myotubes and that this regulation is lost in the absence of functional caveolae as shown in Cav3 P28L and R26Q mutant myotubes and in WT myotubes depleted for Cav3.

**WT Cav3 expression rescue a normal phenotype in Cav3 P28L myotubes**. Our experiments showing that the depletion of Cav3 in WT myotubes faithfully reproduces the mechanoprotection and signaling defects observed in P28L and R26Q myotubes, implies that the absence of Cav3 at the plasma membrane, as a result of its abnormal retention in the Golgi complex, is responsible for the observed phenotype. To validate this hypothesis, we generated stable WT and P28L myoblasts transduced either by GFP alone or by WT Cav3 tagged with GFP (Cav3-GFP). Immunofluorescent microscopy confirmed that expressed Cav3-GFP was mainly localized at the plasma membrane and not retained at the Golgi complex in Cav3-GFP P28L myotubes (Fig. 5a and Supplementary Fig. 4a). We performed electron microscopy to see whether Cav3 WT expression would allow us to reconstitute a pool of structurally defined caveolae at the plasma membrane of Cav3 P28L myotubes expressing GFP or Cav3-GFP. While the plasma membrane of control GFP myotubes presented few, often isolated, caveolae

structures, Cav3-GFP rescued myotubes presented a significantly higher number of bona fide caveolae, including larger vacuolar structures with connected caveolae, i.e., rosettes (Fig. 5b, c), as classically observed in WT myotubes (Fig. 1a). These observations confirm that the decrease in the number of caveolae in Cav3 P28L myotubes is a direct consequence of the retention of Cav3 P28L in the Golgi complex.

Next, we examined whether the reconstitution of the caveolae reservoir at the plasma membrane of Cav3 P28L myotubes was sufficient to rescue the regulatory role of caveolae in mechanoprotection and IL6/STAT3 signaling. We therefore monitored the resistance to membrane bursting of GFP- and WT Cav3-GFP expressing P28L myotubes as described above. Notably, Cav3-GFP P28L myotubes showed a strong increase in the resistance to membrane bursting under hypo-osmotic shock as compared to GFP P28L myotubes (GFP: 49 ± 3%; Cav3-GFP: 18 ± 2%). It also took a significantly longer time for Cav3-GFP P28L myotubes to burst as compared to GFP P28L myotubes (GFP: 1.6 ± 0.1 min; Cav3-GFP: 2.3 ± 0.2 min) (Fig. 6a, b). WT myotubes expressing Cav3-GFP presented a bursting fraction (27 ± 2%) identical to others control cells such as WT myotubes transfected with siCtrl (Fig. 2f and Supplementary Fig. 4b). Consistently, the bursting times of WT myotubes + GFP and WT + Cav3-GFP myotubes were similar (GFP: 3.4 ± 0.3 min; Cav3-GFP: 3.9 ± 0.2 min). The slightly higher resistance to bursting observed in WT myotubes expressing GFP only may be due to the higher level of cell differentiation that we observed in these cells (mean number of nuclei/cell: GFP = 6.45 ± 0.25; Cav3-GFP = 4.1 ± 0.12).

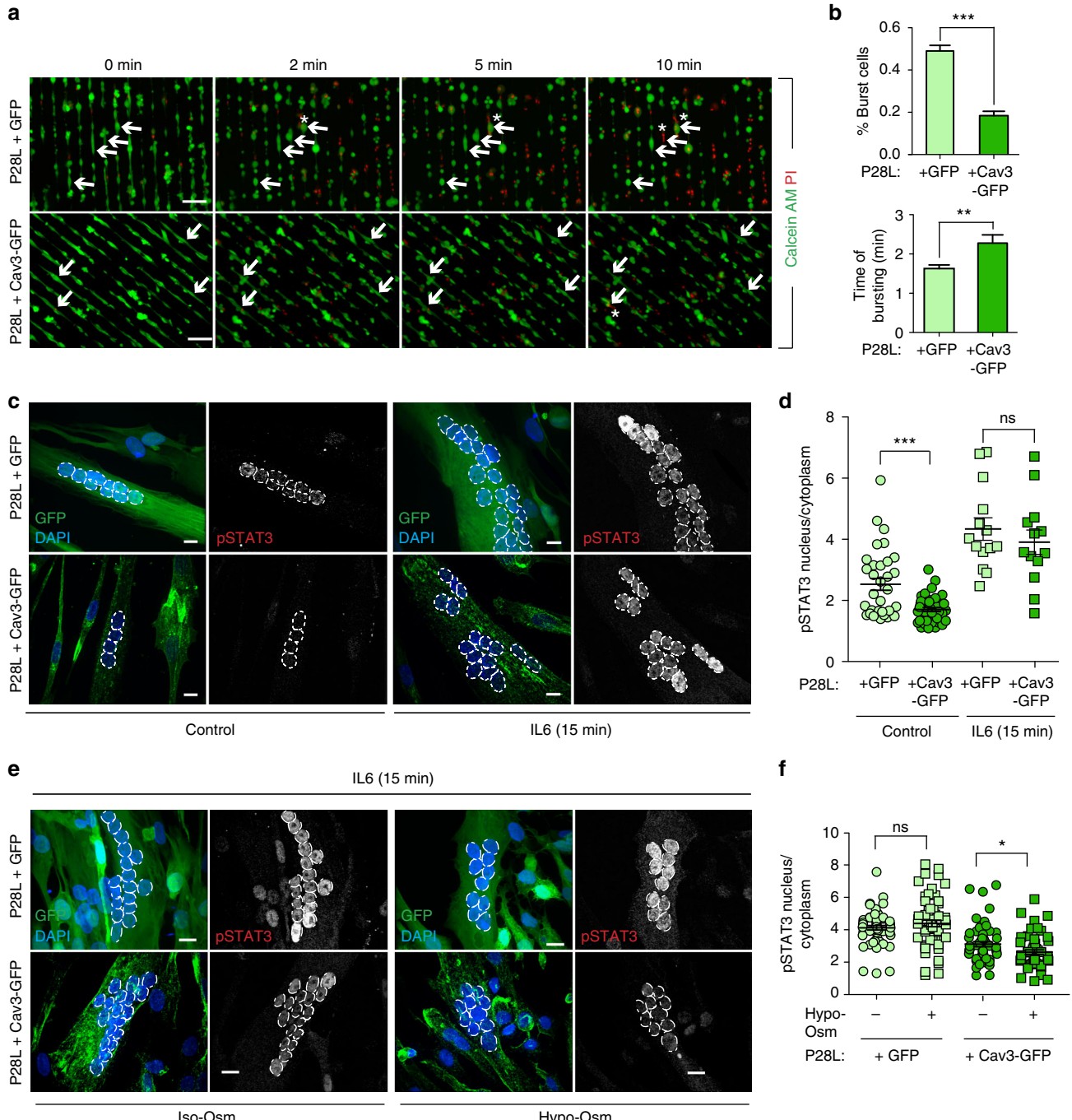

**Fig. 6** Expression of WT Cav3 rescues a normal phenotype in Cav3 P28L myotubes. **a** Micropatterned P28L GFP and P28L Cav3-GFP transduced myotubes were loaded with calcein-AM (green). The medium was switched with a 30 mOsm medium supplemented with propidium iodide (PI, red). Representative pictures were taken at the indicated times during hypo-osmotic shock. Arrows correspond to myotubes and asterisks correspond to burst myotubes. **b** Quantification of the percentage of burst myotubes (upper panel) and mean time of bursting in minutes (lower panel) in **a**. **c** Confocal microscopy of immunofluorescent pSTAT3 in P28L GFP or P28L Cav3-GFP transduced myotubes stimulated or not with 10 ng mL$^{-1}$ IL6 for 15 min. White dashed lines outline nucleus boundaries. **d** Quantification of pSTAT3 nuclear translocation in **c** corresponding to nuclei/cytoplasm mean intensity ratio of pSTAT3. **e** Confocal microscopy of immunofluorescent pSTAT3 in P28L GFP or P28L Cav3-GFP transduced myotubes subjected or not to a 75 mOsm hypo-osmotic shock (Hypo-Osm) for 5 min, followed by stimulation with 10 ng mL$^{-1}$ IL6 for 15 min. White dashed lines outline nucleus boundaries. **f** Quantification of pSTAT3 nuclear translocation in **e** corresponding to nuclei/cytoplasm mean intensity ratio of pSTAT3. **a** Scale bar = 120 μm. **c**, **e** Scale bar = 10 μm. Reproducibility of experiments: **a** Shows representative data of 3 experiments quantified in **b** (% burst cells: GFP $n = 353$ cells and Cav3-GFP $n = 358$ cells; time of burst: GFP $n = 175$ cells and Cav3-GFP $n = 65$ cells). **c** Shows representative data of 3 experiments quantified in **d** (control: GFP $n = 33$ cells and Cav3-GFP $n = 42$ cells; 15 min: GFP $n = 14$ cells and Cav3-GFP $n = 13$ cells). **e** Representative data of 3 experiments quantified in **f** (Iso-Osm: P28L GFP $n = 58$ cells, P28L Cav3-GFP $n = 60$ cells; Hypo-Osm: P28L GFP $n = 57$ cells, P28L Cav3-GFP $n = 52$ cells). Mean value ± SEM. **b** Statistical analysis with two-tailed paired $t$ test. **d**, **f** Statistical analysis with one-way ANOVA; *$P < 0.05$; **$P < 0.01$; ***$P < 0.001$; ns non-significant

Finally, we analyzed the regulation of the IL6/STAT3 pathway by monitoring STAT3 phosphorylation and nuclear translocation in GFP and Cav3-GFP expressing P28L myotubes. At steady state, we observed a significant decrease of pSTAT3 activation and nuclear translocation in Cav3-GFP P28L myotubes as compared to GFP P28L myotubes, indicating that the expression of Cav3 was sufficient to reduce the hyperactivation of STAT3 observed at steady state in Cav3 P28L myotubes (Fig. 6c, d). Upon IL6 stimulation, we measured a slightly decreased but not significant pSTAT3 nuclear translocation in Cav3-GFP P28L myotubes, similarly to what we observed when comparing WT to non-transduced Cav3 P28L myotubes (Figs. 3d and 6d). We also found that the expression of Cav3-GFP in WT myotubes led to a decrease of pSTAT3 activation and nuclear translocation at steady state as compared to non-transduced WT myotubes (Supplementary Fig. 4c). Importantly, the expression of Cav3-GFP in Cav3-P28L mutant myotubes led to a significant decrease of IL6-induced pSTAT3 nuclear translocation under hypo-osmotic conditions in contrast to Cav3-P28L myotubes expressing GFP only (Fig. 6e, f). Altogether, these results show that Cav3 expression allows its localization at the plasma membrane of P28L myotubes, which is sufficient to restore membrane mechanoprotection and the regulation of IL6/STAT3 signaling by caveolae mechanics.

## Discussion

In the present work, we investigated two aspects of caveolae that have been so far poorly characterized in human muscle cells. We first addressed the role of caveolae in mechanosensing and mechanoprotection, a new function of caveolae that has been recently established by several investigators in various cell types. When cells experience acute mechanical stress such as cell swelling or cell stretching, caveolae flatten out into the plasma membrane to provide extra membrane area and prevents membrane tension increase and membrane rupture[14–18]. In agreement with these studies, we found that the presence of functional caveolae was absolutely required to protect human myotubes against severe mechanical stress. Thus, the Cav3 P28L and R26Q mutant myotubes presented a major defect in mechanoprotection with a lack of membrane tension buffering and increased sensitivity to membrane rupture. Whereas the mutant myotubes showed a dramatic decrease in the number of caveolae present at the plasma membrane, the expression of wild type Cav3 allowed to restore a number of caveolae sufficient to restore mechanoprotection. The depletion of Cav3 in healthy myotubes reproduced the phenotypes observed in Cav3 P28L and R26Q myotubes indicating that the retention of Cav3 in the Golgi complex is responsible for the absence of a functional reservoir of caveolae at the plasma membrane and thereby the lack of mechanoprotection in these mutants. Our finding that Cav3 P28L and R26Q myotubes still express caveolae at the plasma membrane, albeit to a much lesser extent and with abnormal morphology, most likely indicates that this number is too low to assure an efficient mechanoprotection and/or that these caveolae are not fully functional as suggested by their aberrant size.

It is tempting to speculate that the increased fragility of the mutant myotubes membrane could be related to the pathological phenotype reported in Cav3-related muscle dystrophies. We were however surprised that these defects were mainly observed when mutant myotubes were subjected to a severe hypo-osmotic shock. Mild hypo-osmotic shocks did not allow to reveal mechanoprotection defects in mutant myotubes. This is indeed in agreement with early electron microscopy studies showing that *Aplysia californica* smooth muscle and frog skeletal muscle fibers must be stretched up to nonphysiological levels such as three

times the in situ length in order to visualize the presence of flattened caveolae[37,38]. This is also consistent with the mild-to-moderate clinical symptoms described in these patients[39]. We further investigated this aspect with in vivo experiments in mice (see Supplementary Methods). We transplanted WT GFP, P28L GFP, or P28L Cav3-GFP transduced human myoblasts in severely cryo-damaged *tibialis anterior* muscles (Supplementary Fig. 5). For the three transplanted cell types, we observed mature muscle fibers containing identical numbers of human cells with nuclei positive for human lamin A/C (Supplementary Fig. 5a, b) and expressing human spectrin (Supplementary Fig. 5c). The localization of human nuclei in the fibers was not affected either (Supplementary Fig. 5d). These results indicate that the Cav3-P28L mutated myoblasts maintain their potential to differentiate in vivo after transplantation into regenerating muscles of mice. Similar results were reported for several muscular dystrophies including Duchenne and facioscapulohumeral muscular dystrophies that present more severe clinical disorders than the Cav3-related muscle dystrophies studied here[40]. Importantly, our in vivo experiments confirmed the rescue results obtained in vitro (Fig. 5b, c) since Cav3-GFP was located at the plasma membrane after fusion of Cav3-GFP-rescued P28L myoblasts with mouse fibers (Supplementary Fig. 5a).

This incited us to explore other functions of caveolae that could also be deregulated by caveolae dysfunction in Cav3 mutant myotubes. We investigated the regulation of muscle cells signaling as several signaling defects have been described in muscle dystrophies and caveolae have long been associated with the regulation of intracellular signaling. Indeed, Cav3 has been involved in the regulation of distinct signaling pathways important for muscle function such as calcium homeostasis[41], the insulin/GLUT4/Akt pathway[42] or TrkA and EGFR signaling[26]. We focused our analysis on the interleukin-6 (IL6)/STAT3 signaling pathway which has been shown to play an essential role in muscle tissue homeostasis[43]. In addition, the IL6 pathway is tightly associated to mechanical stress in muscle cells as IL6 is secreted mostly during physical exercise[44]. IL-6 ligand expression was also shown to be triggered by long-term osmotic variations in non-muscle cells such as corneal epithelial cells and Caco-2 cells[45,46].

Our data show a major deregulation of the IL6/STAT3 signaling pathway in the Cav3 mutant myotubes with a constitutive hyperactivation of STAT3 at steady state. In comparison to WT myotubes, we found no significant difference in IL6Rα and gp130 total expression, the two subunits of the IL6 receptor, suggesting that the machinery of the IL6 signaling pathway is not modified in those myotubes (Supplementary Fig. 6a). Interestingly, a recent study reported that Cavin-1, a constituent of caveolae, was required for the proper localization of SOCS3 in endothelial cells at rest, and that STAT3 activation was increased when Cavin-1 was depleted[47]. Although more investigations are required, this regulation is unlikely to occur in Cav3 P28L and Cav3 R26Q mutant myotubes as they express similar levels of Cavin-1, and high levels of SOCS3 as expected from Fig. 3g (Supplementary Fig. 6a).

The deregulation of the IL6/STAT3 signaling pathway translated into increased STAT3 nuclear translocation and expression of *MYH8*, *SOCS3*, *ACTC1*, and *ACTN2*, genes that are known to be regulated by STAT3 and that have been associated with muscle development and regeneration. As for the defects in mechanoprotection, the deregulation of the IL6/STAT3 signaling pathway could be reproduced by depleting healthy myotubes from Cav3, indicating that the absence of Cav-3 and/or caveolae was responsible for the hyperactivation of STAT3. More importantly, we observed that IL6/STAT3 signaling was regulated by mechanical stress in a Cav3-dependent manner in human

myotubes. The regulation of IL6/STAT3 mechanosignaling by caveolae was lost in Cav3-mutant myotubes or when healthy myotubes were depleted of Cav3. Again, as observed for mechanoprotection, the regulation of IL6 mechanosignaling could be rescued in P28L Cav3 myotubes transduced by the WT form of Cav3, supporting the role of bona fide caveolae in the regulation of these two processes.

The first CAV3 mutation associated with muscle disorders was described 20 years ago and today it has been extended to five distinct genetic disorders: rippling muscle disease (RMD), distal myopathies (DM), hyperCKemia (HCK), limb-girdle muscular dystrophy 1C (LGMD-1C), and familial hypertrophic cardio-myopathy (HCM)[20,39]. Although many studies have addressed the role of these mutations in muscle damages, the underlying mechanisms remain poorly characterized. Cav3 has been involved in several aspects of muscle physiology including myoblast fusion[48] and T-tubules organization[49]. Moreover, Cav3 interacts with the dystrophin complex[50] and regulates the trafficking of dysferlin[51], two important muscle proteins whose expression and localization are deregulated in severe myopathies. It is therefore likely that the mechanisms by which Cav3 mutations are responsible for muscle dystrophies are multiple. Several caveolin–protein interactions including signaling molecules such as endothelial nitric oxide synthase (eNOS) or PTEN have been proposed to occur through the caveolin scaffolding domain (CSD)[52], a 20 amino acid region within Cav1 and Cav3. We found that cavtratin, a CSD mimicking peptide[53] was able to significantly reduce the hyperactivation of the IL6/STAT3 path-way in Cav3 P28L mutant myotubes (Supplementary Fig. 6c, d). Together with the absence of the IL6R subunit and gp130 into caveolae (Supplementary Fig. 6b), our data suggest that the reg-ulation of IL6 mechanosignaling requires the interaction with Cav3-CSD outside of caveolae. These results may imply that other CSD-mediated signaling interactions could be mediated by caveolae mechanics in muscle cells. In this context, it is inter-esting that the Cav3 R26Q mutant myotubes demonstrated a significant increase in NO production[22]. Finally, proteomics performed in CAV3 p.P104L transgenic mice, where Cav3 is also retained at the Golgi apparatus, revealed significant changes in protein expression including signaling molecules[54].

In conclusion, we describe a mechanism by which the Cav3 mutations can be deleterious in human myotubes. We uncovered a new regulation of IL6/STAT3 signaling by caveolae under mechanical stress. Our findings revealed a striking similarity between the regulation of mechanoprotection and the control of IL6/STAT3 signaling by caveolae under mechanical stress. Our data confirm that the retention of Cav3 P28L and R26Q in the Golgi complex is responsible for the absence of functional and morphologically defined caveolae at the plasma membrane, which in turn results in deficient mechanoprotection and IL6/STAT3 mechanosignaling. The IL6/STAT3 pathway is tightly associated with the regulation of muscle mass and size[31,43]. It is likely that the alteration of mechanoprotection and muscle size, two critical parameters for general muscle homeostasis, are deleterious for muscle tissue integrity. It is therefore tempting to propose that the caveolae-dependent mechano-regulation of the IL6/STAT3 pathway that we have unraveled here is critical to couple the activation of the IL6/STAT3 pathway with the intensity of mechanical stress that myotubes constantly experience during their lifetime thereby preventing a chronic hyperactivation of IL6/STAT3, through a negative feedback loop, that would be other-wise pathological to muscle cells.

## Methods

**Cell lines**. Human primary cells were obtained from EuroBioBank. P28L and R26Q human myoblasts were immortalized by the platform for immortalization of human cells of the Institute of Myology[40]. Briefly, myoblasts were transduced with lentiviral vectors encoding hTERT and cdk4 and containing puromycin (P28L) or puromycin and neomycin (R26Q) selection markers. Transduced cells were selected with puromycin (1 µg mL$^{-1}$) for 6 days (P28L) or with puromycin (1 µg mL$^{-1}$) for 6 days and neomycin (1 mg mL$^{-1}$) for 10 days (R26Q). Cells were seeded at clonal density, and individual myogenic clones were isolated. For caveolin-3 expression, immortalized WT and P28L myoblasts were transduced with lentiviral vectors expressing WT caveolin-3 and a GFP reporter gene (MOI 5). A GFP lentiviral vector was used as control (MOI 5).

**Cell culture**. All cells were grown at 37 °C under 5% of CO$_2$. All myoblasts cell lines were cultured in Skeletal Muscle Cell Growth Medium (Promocell) supplemented with 20% FCS (Gibco, Life Technologies), 50 µg mL$^{-1}$ of fetuine, 10 ng mL$^{-1}$ of epidermal growth factor, 1 ng mL$^{-1}$ basic fibroblast growth factor, 10 µg mL$^{-1}$ of insulin, and 0.4 µg mL$^{-1}$ of dexamethasone (Promocell). Prior to any cell seeding, surfaces (well, coverslip, patterned coverslips) are coated with 1% of matrigel (v/v) (Sigma) for 15 min at 37 °C. For myoblast differentiation, confluent cells (80–100% confluency) are put in DMEM high-glucose Glutamax (Gibco, Life Technologies), supplemented with 0.1% of insulin (v/v) (Sigma) for 4 days.

**Antibodies and reagents**. Mouse anti-αTubulin (Sigma-Aldrich, clone B512, T5168, 1/1000 for WB); mouse anti-caveolin-3 (Santa Cruz, clone A3, sc-5310, 1/1000 for WB, 1/250 for IF); rabbit anti-caveolin-1 (Cell Signaling, 3238, 1/1000 for WB, 1/500 for IF); goat anti-GM130 (Santa Cruz, clone P-20, sc-16268, 1/50 for IF); mouse anti-MF20 (myosin heavy chain) (kind gift of Vincent Mouly, 1/100 for WB, 1/20 for IF); mouse anti-STAT3 (Cell Signaling, clone 124H6, 9139, 1/1000 for WB); rabbit anti-pSTAT3 (Cell Signaling, clone D3A7, 9145, 1/1000 for WB, 1/75 for IF); Secondary antibodies conjugated to Alexa FITC, Cy3, Cy5 (1/200 for IF) or horseradish peroxidase (1/1000 for WB) (Beckman Coulter or Invitrogen). DAPI (Sigma-Aldrich).

**RNA interference-mediated silencing**. Myoblasts were transfected with small interfering RNAs (siRNAs) using HiPerFect (Qiagen) according to the manu-facturer's instructions at days 0 and 2 of differentiation and were cultured in differentiation medium for a total of 4 days. Experiments were performed on the validation of silencing efficiency by immunoblot analysis using specific antibodies and normalizing to the total level of tubulin used as loading controls. 20 nM of a pool of four siRNA targeting Cav3 were used (SI03068730, SI02625665, SI02625658, and SI00146188, QIAGEN), Control siRNA (1022076, QIAGEN) was used at the same concentration and served as a reference point (Supplementary Table 1).

**Immunoblotting**. Cells were lysed in sample buffer (62.5 mM Tris/HCl, pH 6.0, 2% v/v SDS, 10% glycerol v/v, 40 mM dithiothreitol, and 0.03% w/v phenol red). Lysates were analyzed by SDS-PAGE and Western blot analysis and immuno-blotted with the indicated primary antibodies and horseradish peroxidase-conjugated secondary antibodies. Chemiluminescence signal was revealed using Pierce™ ECL Western Blotting Substrate, SuperSignal West Dura Extended Dura-tion Substrate or SuperSignal West Femto Substrate (Thermo Scientific Life Technologies). Acquisition and quantification were performed with the ChemiDoc MP Imaging System (Bio-Rad) (Supplementary Fig. 7).

**Immunofluorescence**. Myoblasts were grown and differentiated on coverslips for 4 days. For Cav3, Cav1, MF20, GM130 staining, cells are fixed with 4% PFA (v/v) (Sigma-Aldrich) for 10 min at RT, quenched in 50 mM NH$_4$Cl and then per-meabilized with 0.2% BSA (v/v) and 0.05% saponin (v/v) (Sigma-Aldrich) in PBS for 20 min. Cells are incubated sequentially with indicated primary and fluorescence-conjugated secondary antibody in permeabilization buffer for 1 h at RT. For pSTAT3 staining, cells are fixed and permeabilized with cold methanol for 15 min at −20 °C. After washes with PBS 0.2% BSA (v/v), cells are incubated sequentially with indicated primary and fluorescence-conjugated secondary anti-body in PBS 0.2% (v/v) for 1 h at RT. In both protocols, coverslips are mounted in Fluoromount-G mounting medium (eBioscience) supplemented with 2 µg mL$^{-1}$ of DAPI (Sigma-Aldrich). Acquisitions of images are done using a spinning disk microscope (inverted Spinning Disk Confocal Roper/Nikon; Camera: CCD 1392 × 1040 CoolSnap HQ2; objective: ×60 CFI Plan Apo VC). Colocalization between markers was quantified by the Manders' coefficient using ImageJ software colo-calization plugin JACoP[55].

**Electron microscopy**. Epon embedding was used to preserve the integrity of cell structures. Myotubes were fixed sequentially for 1 h at room temperature with 1.25% glutaraldehyde in 0.1 M Na-Cacodylate and then overnight at 4 °C.

Cells were washed extensively with 0.1 M Na-Cacodylate, pH 7.2. Membrane fixation was performed for 1 h at room temperature with 1% OsO$_4$ in 0.1 M Na-Cacodylate, pH 7.2. Cells were dehydrated by incubation with aqueous solutions of ethanol at increasing concentrations (50, 70, 90, then 100%, each for 10 min at RT). Embedding was finally performed in LX112 resin. Cells were infiltrated with a 1:1 LX112:ethanol solution, washed with LX112, and embedded overnight at 60 °C in

LX112 resin. Ultrathin 65 nm sections were sliced using a Leica UCT ultramicrotome and mounted on nickel formvar/carbon-coated grids for observations. Contrast was obtained by incubation of the sections for 10 min in 4% uranyl acetate followed by 1 min in lead citrate.

Electron micrographs were acquired on a Tecnai Spirit electron microscope (FEI, Eindhoven, The Netherlands) equipped with a 4k CCD camera (EMSIS GmbH, Münster, Germany)

**Micropatterning**. 18 mm coverslips were micropatterned as described previously[56] using a photo-mask with lines of 10 μm of width, separated by 60 μm. In both force measurements and membrane bursting assay, myoblasts are plated at confluency on line micropatterns coverslips coated with 1% of matrigel (v/v) (Sigma) for 15 min at 37 °C. Differentiation of myoblasts is achieved as described above in section "Cell culture".

**Force measurements**. Plasma membrane tethers were extracted from cells by a concanavalin A (Sigma-Aldrich) coated bead (3 μm in diameter, Polysciences) trapped in optical tweezers. The optical tweezers are made of a 1064 nm laser beam (ytterbium fiber laser, $\lambda = 1064$ nm, TEM 00, 5 W, IPG Photonics, Oxford, MA) expanded and steered (optics Elliot Scientific, Harpenden, UK) in the back focal plane of the microscope objective (Apo-TIRF ×100 NA 1.45, Nikon). The whole setup was mounted on a Nikon Eclipse-Ti inverted microscope. The sample was illuminated by transmitted light, and movies were acquired at 10 Hz with an EM-charge-coupled device camera (Andor iXon 897) driven by Micro-Manager. The fine movements and particularly the translational movement necessary to pull the membrane tether were performed using a custom-made stage mounted on a piezoelectric element (P753, Physik Instrumente, Karlsruhe, Germany) driven by a servo controller (E665, Physik Instrumente) and a function generator (Sony Tektronix AFG320).

Calibration was performed using an oscillatory modulation driven by a function generator and measuring the response of the bead to an oscillatory motion of the stage. We measured $k = 159$ pN μm$^{-1}$. This relationship is linear in the laser power range used for the experiments (0.4–1.2 W).

The membrane tether was held at constant length to measure the static force. For measuring membrane tension changes due to hypo-osmotic shock, a first tether was first pulled at 300 mOsm (iso condition). A second tube was pulled on the same cell 5 min after diluting the medium with MilliQ water to obtain 45 mOsm. The position of the bead used to compute tether forces was detected from the images using a custom ImageJ macro.

**Membrane bursting assay**. Line micropatterned myotubes are incubated in 5 μg mL$^{-1}$ of calcein-AM (Life Technologies) and 50 μg mL$^{-1}$ of DAPI (Sigma-Aldrich) for 15 min at 37 °C in the dark. Medium was then switched back with differentiation medium to wash out the excess of calcein-AM. The medium is then switched again with a 30 mOsm hypo-osmotic shock medium obtained after a dilution of 10% medium and 90% H$_2$O, supplemented with 2 mg mL$^{-1}$ of PI (Sigma). Immediately after medium switching, pictures are taken every minute for 10 min using a video microscope (Inverted microscope Nikon Ti-E, Camera: CCD 1392 × 1040 CoolSnap HQ2, objective: ×10 CFI Fluor). The time of bursting corresponds to the mean of the bursting time (i.e., the first frame where PI appears in the nuclei) for each burst myotube in a given condition.

**IL6 stimulation**. Myotubes were starved 4 h by switching the differentiation medium to DMEM medium. In resting conditions, cells are then stimulated by switching the medium with DMEM with 0.2% BSA (w/v), supplemented with 10 ng mL$^{-1}$ of human recombinant IL6 (R&D) for 0, 5, or 15 min at 37 °C. For hypo-osmotic conditions, medium was first switched to 75% hypo-osmotic shock (25% DMEM, 75% H$_2$O) for 5 min and then switched to the same medium supplemented with 10 ng mL$^{-1}$ of IL6 for 5 more minutes at 37 °C. For stretching conditions, myoblasts were differentiated on fibronectin (Sigma-Aldrich) coated stretchable plates (Uniflex® culture plate, Flexcell International) and were then subjected or not to 30 min of cyclic stretch (10% elongation, 0.5 Hz), using the FX-4000T TM Tension Plus device (Flexcell International), followed or not by 5 min of 10 ng mL$^{-1}$ IL6 stimulation. Cells are lysed and samples are analyzed by immunoblotting. For the analysis, pSTAT3 levels were quantified by calculating the ratio between pSTAT3 and STAT3, both normalized to Tubulin signal as follows: (pSTAT/tubulin$^{\text{pSTAT}}$)/(STAT/tubulin$^{\text{STAT}}$). For the analysis of pSTAT3 nuclear translocation, myotubes were differentiated on coverslips, stimulated with IL6 as described above for 0 or 15 min and were then fixed for further immunofluorescence analysis. Quantification corresponds to the ratio between the nuclei and the cytoplasm of the mean pSTAT3 intensity.

**Quantitative PCR**. Cells were lysed and RNA extraction was performed using an extraction kit (RNeasy Plus, Qiagen). Reverse-transcription reaction was performed with 1 μg of RNA per reaction, using high capacity cDNA reverse-transcription kit (Applied Biosystem). qPCR was performed on 50 ng of cDNA for a reaction in a total volume of 20 μL, using TaqMan Gene Expression Assays (GAPDH: Hs02786624_g1; ACTC1: Hs01109515_m1; MYH8: Hs00267293_m1; SOCS3: Hs02330328_s1, ACTN2: Hs00153809_m1, Applied Biosystem) and a Lightcycler

480 Probes Master kit (Roche). Relative expression levels were calculated using ΔΔCT method with fold changes calculated as $2^{-\Delta\Delta\text{CT}}$.

**Statistical analyses**. All analyses were performed using GraphPad Prism version 6.0 and 7.0, GraphPad Software, La Jolla, CA, USA, www.graphpad.com. Two-tailed (paired or unpaired) $t$-test was used if comparing only two conditions. For comparing more than two conditions, Kruskal–Wallis test was used with Dunn's multiple comparison test (if comparing all conditions to the control condition). Significance of mean comparison is marked on the graphs by asterisks. Error bars denote SEM or SD.

**Reporting summary**. Further information on experimental design is available in the Nature Research Reporting Summary linked to this article.

## Data availability

The authors declare that the main data supporting the findings of this study are available within the article and its Supplementary Information files. Extra data are available from the corresponding authors upon request.

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

## Acknowledgements

The authors would like to thank Catherine Coirault, Stéphane Vassilopoulos, Nicolas Carpi, and the laboratory of Bruno Goud for providing materials and/or expertise. The authors are grateful to Paolo Pierobon for help on the analysis of membrane tension measurements. The authors thank the platform for immortalization of human cells of the Institute of Myology and Jessica Ohana for her help with histological analysis. The authors wish to thank Alison Forrester for carefully reading the manuscript. The facilities as well as scientific and technical assistance from staff in the PICT-IBiSA/Nikon Imaging Centre at Institut Curie-CNRS and the France-BioImaging infrastructure (No. ANR-10-INSB-04) are acknowledged. The electron microscope facility was supported by the French National Research Agency through the "Investments for the Future" program (France-BioImaging, ANR-10-INSB-04). This work was supported by institutional grants from the Curie Institute, INSERM, and CNRS, and by specific grants from Association Française contre les Myopathies (AFM): CAV-MUT (17151) to M.D.; CAV-STRESS-MUS (14266 to C.M.B. and 14293 to C.L.), and Agence Nationale de la Recherche (DECAV-RECAV No. ANR-14-CE09-0008-03 to C.L.). J.P. was funded by Polish Ministry of Science and Higher Education Mobility Plus program (1668/MOB/V/2017/0). The Johannes and Lamaze teams, the PICT-IBiSA/Nikon Imaging Centre at Institut Curie-CNRS, and the France-BioImaging infrastructure are members of Labex CelTisPhyBio (No. ANR-10-LBX-0038) and of IDEX PSL (No. ANR-10-IDEX-0001-02 PSL).

## Author contributions

M.D. designed and performed the experiments, analyzed results, and wrote the manuscript. D.K., B.S., C.V.L., V.C., M.B., E.N. and J.P. performed experiments or analysis. A.B., N.T., L.J., P.N. and G.B.-B. provided technical support and conceptual advice. C.L. and C.M.B. supervised the project, designed experiments, and wrote the manuscript.

## Additional information

**Competing interests:** The authors declare no competing interests.

