## [Peer Review File · Nature Communications]

Reviewers' comments:

Reviewer #1 (Remarks to the Author):

- What are the major claims of the paper?

Dewulf et al. show that caveolin-3 mutations

- 1) lead to a reduced number of caveolae in myotubes, because most of the caveolin-3 mutant protein gets stuck in the Golgi
- 2) reduce the elasticity/expansibility and tear resistance of the plasma membrane
- 3) induce constitutive IL-6 signalling as demonstrated by STAT3 tyrosine phosphorylation, STAT3 nuclear localization and STAT3-dependent gene induction

In sum the authors suggest that mechanical stress (by hypoosmolarity, training) hinders IL-6 signalling. This interference depends on Caveolin-3.

- Are the claims novel? If not, please identify the major papers that compromise novelty

The authors investigate the functional relevance of two Caveolin-3 mutants (P28L and R26Q), their impact on plasma membrane stability and IL-6 signalling. The major new aspects of this study are the mutants which are very intensively compared with wild type Caveolin-3. The impact of caveolae and Caveolin-3 for the stability of myotubes and the impact of caveolae on IL-6 signalling is known from other studies (e.g. PMID: 11431690; PMID: 29330478 respectively). Several other studies addressed IL-6 signalling induced by hyperosmolarity (e.g. PMID: 10514514).

- Will the paper be of interest to others in the field?

Mainly interesting for scientists working with these specific Caveolin-3 mutants. Although the principal subject is not totally new, the paper is of some interest to others in the field (cell biologists) as the function of caveolae is still under debate.

- Will the paper influence thinking in the field?

The impact of caveolae on membrane flexibility and signalling is interesting. However, it remains to be shown that the effect of Caveolin-3 on IL-6 signalling is specific and does not affect all other signalling pathways as well.

- Are the claims convincing? If not, what further evidence is needed?

Claims and interpretations are correct. Just a few minor (technical, editorial) points need to be addressed:

Numbering in figure 1 and in the legend to figure 1 does not correlate. (d should be e, c should be d etc.)

Figure 2d and f: WT cells and siCtl cells strongly differ in respect to % burst cells and time of bursting, despite the fact that both are control cells. Reliability of the assay?

Figure 3a and 4a: STAT3 expression is strongly increased in Caveolin-3 P28L expressing cells. Please comment!

Figures 3a,b,e,f, Figure 4, S3:

Obviously, STAT3 and (p)STAT3 staining was not from the same gel (no stripping of the membrane) but from two gels running in parallel. Thus, each gel has its own tubulin staining for quantification. Where is the 2nd tubulin staining?

$(\text{pSTAT3}/\text{tub}) / (\text{STAT3}/\text{tub}) = (\text{pSTAT3}/\text{STAT3})$ if only one tubulin staining has been considered!

In the case of two separate gels for (p)STAT and STAT calculation should be:

$(\text{pSTAT3}/\text{tub}^{\text{(pSTAT)}}) / (\text{STAT3}/\text{tub}^{\text{(STAT)}})$ because the tubulin signal from the pSTAT-gel differs from the tubulin signal on the STAT-gel.

In case only one gel was used and been stripped after (p)STAT staining, $(\text{pSTAT3}/\text{tub}) / (\text{STAT3}/\text{tub}) = (\text{pSTAT3}/\text{STAT3})$ and tubulin signals are meaningless.

Figure S1 c: please indicate in the figure legend that the figures correspond to the identical figures shown in figure 1e.

- Are there other experiments that would strengthen the paper further? How much would they improve it, and how difficult are they likely to be?

Studying the mode of inhibition of IL-6 signalling would be interesting, but not absolutely necessary. (receptor internalization, SOCS3-mediated inhibition, phosphatases ...?)

- Are the claims appropriately discussed in the context of previous literature?

There are some more data available on osmo-sensing and IL-6 signalling, caveolae and IL-6 sensing, and caveolae and membrane tension. However, the most important studies have been considered.

- If the manuscript is unacceptable in its present form, does the study seem sufficiently promising that the authors should be encouraged to consider a resubmission in the future?

Only very minor points need to be corrected.

- Is the manuscript clearly written? If not, how could it be made more accessible?

The paper is very well written and easy to read. Presentation of data is very clear.

- Could the manuscript be shortened to aid communication of the most important findings?
No.

- Have the authors done themselves justice without overselling their claims?

No

- Have they been fair in their treatment of previous literature?

See above

- Have they provided sufficient methodological detail that the experiments could be reproduced?

Yes, but some figure legend could be improved. Some stainings are missing (see above)

- Is the statistical analysis of the data sound?

Yes (see above)

- Should the authors be asked to provide further data or methodological information to help others replicate their work? (Such data might include source code for modelling studies, detailed protocols or mathematical derivations).

no

- Are there any special ethical concerns arising from the use of animals or human subjects?

No concerns in this respect.

Reviewer #2 (Remarks to the Author):

In this manuscript, the authors examine the effect of caveolae on the response of myotubes to mechanical stress. They study myotubes expressing either of two non-functional caveolin-3 mutants, associated with different muscle diseases, and also suppress caveolin-3 expression in wild-type myotubes with siRNA.

The technical quality of the experiments is very good, and (with the caveat detailed below), the results are believable. The main issue for publication lies in the significance of the findings.

The authors have already published the most important results shown in Figs. 1 and 2. Specifically, in their seminal 2011 Cell paper (the first report that caveolae protect muscle cells against mechanical stress), the authors showed that myotubes expressing Cav3 P28L, which is retained in the Golgi, fail to buffer the rise in membrane tension in response to mechanical stress, and that they rupture more easily than normal myotubes during hypo-osmotic shock. The new findings shown in Figs. 1 and 2 are that cells expressing Cav3-P28L and Cav3-R26Q have reduced levels of cell-surface caveolae, and that Cav3-R26Q (also retained in the Golgi) has the same effects as Cav3-P28L. These are expected results.

Figure 5 shows that the defects in Cav3-P28L myotubes can be overcome by expression of WT Cav3-GFP. This is an important control, but in itself does not contribute important new information.

The most significant results are shown in Figures 3 and 4. Myotubes normally respond to IL6 by increasing phosphorylation of STAT3. The authors show that myotubes expressing the Cav3 mutants have constitutively higher levels of pSTAT3 than WT, and that the IL6-induced increase in pSTAT3 levels is further enhanced in the mutants, and also in siCav3 myotubes. Four transcriptional targets of pSTAT3 are also constitutively up-regulated in the mutants. This is the first demonstration that IL6 signaling is negatively regulated by caveolae in muscle cells. Given the important role of IL6 signaling in muscle physiology and disease, this is a novel and significant

result, and probably justifies publication.

This finding is extended in Figure 4 and Supplementary Figure 3. The authors first present evidence that prior mechanical stress (hypo-osmotic shock or stretch) inhibits subsequent STAT3 phosphorylation in response to IL6 treatment in normal myotubes. Next, they show that this inhibitory effect is lost in myotubes lacking caveolae (Cav3 mutants or Cav3 siRNA). While interesting, the significance of these findings is tempered a bit by the quality of the data. I am not aware of any prior demonstration that mechanical stress normally inhibits the IL6 response in muscle. Figures 4a and 4b show pSTAT3 levels in WT myotubes after 0 or 5 min of IL6 treatment, with or without prior hypo-osmotic shock. The authors report a robust 80% inhibition in IL6-mediated stimulation of STAT3 phosphorylation by hypo-osmotic shock (though all the relevant bands on the representative blot shown are quite faint).

However, in Figure 4c, they repeat the same experiment in siRNA-treated cells. Myotubes treated with control siRNA (expected to be the same as the control cells in Figure 4a) show a much less impressive (though still significant to $p < 0.05$) inhibition of IL6-induced STAT3 phosphorylation by prior hypo-osmotic shock than the control myotubes in Figure 4a. Together with the overall unevenness of the bands in the blots shown, this discrepancy somewhat reduces the impact of the finding.

Minor points.

1. The authors should fix the legend to Figure 1. They apparently merged the original panels c and d, and eliminated the original panel e, without changing the legend.
2. The authors should comment on why mutant caveolin-3 retains endogenous caveolin-1 in the Golgi (Supp. Fig. 1), but WT-Cav3-GFP overexpressed in mutant cells is transported to the cell surface (Figure 5).
3. Figure 2c and d. They should say how the mean time of bursting was calculated.
4. Total STAT3 levels are much higher in Cav3-P28L myotubes than in WT or Cav3-R26Q myotubes. The authors might comment on this.
5. Quantifications of bands in blots are expressed as (pSTAT3/tubulin)/(STAT3/tubulin). This is mathematically the same as pSTAT3/STAT3, which is simpler and avoids the possible complication of slight spurious variations in tubulin signal between lanes.
6. Fig. 3c and d. Why did the authors quantitate the pSTAT3 nucleus:cytoplasm ratio, instead of the nucleus:cytoplasm ratio of total STAT3? The regulated step in activation is STAT3 phosphorylation, not nuclear import of pSTAT3. Is the OT value for WT meaningful, given the low levels of pSTAT3 ("little tyrosine phosphorylation of STAT3, if any")? They should also state explicitly that the blue stain in Fig. 3c is DAPI.
7. Fig. 3e and f. They should re-write text to tone down the conclusions a bit to accurately reflect the data. There's no significant constitutive activation of STAT3 without IL6 addition in Cav3 siRNA-treated cells. For this reason, the depletion of cav1 in WT myotubes doesn't exactly reproduce the phenotype of the Cav3 mutants. These results do NOT "demonstrate that the absence of Cav3 and/or caveolae at the PM of mutant myotubes is responsible for the constitutive hyperactivation of the IL6/STAT3 signaling pathway".
8. Discussion. Line 12: should be sensitivity, not sensibility. Line 15; should be restore, not reinstall.

Reviewer #3 (Remarks to the Author):

The present work by Dewulf et al investigates the role of Caveolin-3 in mechanosensing of human skeletal muscle cells and regulation of IL-6/STAT3 signaling. The authors show that myotubes derived from patients with P28L and R26Q mutations in the Caveolin-3 gene, which result in multiple human disease manifestations, including hyperCKemia, rippling muscle disease and

neuropathies, exhibit a reduction in the density of caveolae on the surface of the cells. Mutant proteins are mainly retained in the Golgi instead of cell surface localization. Exposure of mutant myotubes to hypo-osmotic shock shows that mutant cells are impaired in their ability to cope with mechanical stress. The authors further link absence of Cav-3 and reduction of caveolae to a dysregulation of IL-6/STAT3 signaling. Mutations in caveolin-3 are associated with a broad spectrum of human diseases, thus understanding the mechanism underlying the effects of these mutations on diseases is relevant. While it has been previously shown that mutant Cav-3 proteins accumulate in the Golgi and lead to a reduced number of caveolae, the novelty of the current work is the link with mechanosensing and the effects on IL-6/STAT3 signaling in human muscle cells. The manuscript is well written. However additional experiments are required in order to fully support the authors' interpretation. Specifically, more mechanistic depth on the link between Cav-3, caveolae and IL-6/STAT3 signaling as well as the potential impact of STAT3 on the observed phenotype should be provided. In addition, it would be useful to provide validation of the in vitro findings in vivo by cell transplantation assays into mice.

Major points:

- 1- Mechanism connecting Cav-3 and caveolae to IL-6/STAT3 signaling: The authors cite previous work in which the gp130 receptor has been shown to localize in caveolae in a myeloma cell line. This should be shown also in differentiated human myotubes. In addition, it is unclear why the absence of Cav-3 or caveolae leads to hyperactivation of the IL-6/STAT3 signaling. Are the IL-6 receptor levels on the surface of the cells increased in the mutants vs wild-type cells? Or is it due to intracellular signaling? Or to direct interaction Cav-3/STAT3 or Cav-3/gp130? Increasing the depth of the molecular mechanism underlying the defect in the IL-6/STAT3 signaling in these human mutant cells would significantly strengthen this work.
- 2- Impact of STAT3 signaling on the observed phenotype: this is an underdeveloped aspect of the current work: is STAT3 signaling involved in the defect in mechanosensing and consequent bursting or is it unrelated?
- 3- All the presented work has been performed in vitro, while it would be useful to also test in transplantation assays whether mutant and healthy human muscle cells behave differently when injected intramuscularly into recipient mice.
- 4- In Figure 3e-f the transient knockdown of Cav3 leads to hyperactivation of STAT3 upon IL-6 stimulation, but not at steady state, while the mutant cells exhibit elevated STAT3 signaling also at steady state. Can the authors comment on this different phenotype?
- 5- The two Cav-3 mutations assessed lead to different disease manifestations, thus it would be useful to include a discussion of how the observed phenotype reported here might contribute to the two different conditions.

Specific Points:

- 1- In Figure 1d and Suppl. Figure 1c, a quantification of the colocalization between Cav-3 and GM130 and Cav-1 in the human cells should be shown, in order to substantiate the authors' interpretation, or the findings remain only qualitative.
- 2- In Figure 3b statistical significance of the difference in STAT3 phosphorylation should be indicated in both figure and figure legend.

Response to reviewers:

We would like to thank the reviewers for their positive comments and their constructive input. In response to their remarks, we have conducted a significant number of additional experiments and have introduced textual changes to the manuscript. The new experiments and changes to the manuscript are summarized below. The reviewers' comments appear in red, our point-to-point responses are in black. All the figure/table numbers are those of the revised manuscript, except Figures R1-R5 that correspond to the figures found only in this rebuttal. The new text in the manuscript is written in blue.

Reviewer #1

• What are the major claims of the paper?

Dewulf et al. show that caveolin-3 mutations:

- 1) lead to a reduced number of caveolae in myotubes, because most of the caveolin-3 mutant protein gets stuck in the Golgi
- 2) reduce the elasticity/expansibility and tear resistance of the plasma membrane
- 3) induce constitutive IL-6 signalling as demonstrated by STAT3 tyrosine phosphorylation, STAT3 nuclear localization and STAT3-dependent gene induction

In sum the authors suggest that mechanical stress (by hypoosmolarity, training) hinders IL-6 signalling. This interference depends on Caveolin-3.

• Are the claims novel? If not, please identify the major papers that compromise novelty

The authors investigate the functional relevance of two Caveolin-3 mutants (P28L and R26Q), their impact on plasma membrane stability and IL-6 signalling. The major new aspects of this study are the mutants which are very intensively compared with wild type Caveolin-3. The impact of caveolae and Caveolin-3 for the stability of myotubes and the impact of caveolae on IL-6 signalling is known from other studies (e.g. PMID: 11431690; PMID: 29330478 respectively). Several other studies addressed IL-6 signalling induced by hyperosmolarity (e.g. PMID: 10514514).

Indeed, Betz et al (Nat Genet 2001 **PMID:11431690**) is a short study reporting the mislocalization of Cav3 and nNOS in muscle sections of a Cav3 A45T/V patient. Mouse C2C12 derived myotubes transfected by R26Q, A45T/V et P104L Cav3 mutants reveal nNOS hyperactivation. No mechanism is provided and the role of mechanical stress has not been investigated.

The reviewer may have mistakenly cited **PMID:10514514** as this study reports the effect of hyperosmolarity on STAT1 activation via the MAPK pathway. We found however two other studies showing that IL-6 ligand expression can be triggered by long-term osmotic variations in corneal epithelial cells (**PMID: 25255138**) and in Caco-2 cells (**PMID: 15388230**). Although these studies do not involve caveolae and muscle, they further associate the IL-6 pathway with mechanosignaling.

We were well aware of the interesting work published by the Palmer and Pilch groups (Williams et al; *Nat Commun* 2018; **PMID: 29330478**). We thought that this work was less relevant to the mechanism reported in our study since it addresses the role of SOCS3, Cavin-1 and Cav1 in endothelial cells under resting conditions. This study reports that Cavin-1 controls the proper localization of SOCS3 and that a defect in this process is associated with increased STAT3

activation. We have now tested whether this mechanism could be associated to the increased activation of STAT3 in Cav3 mutant myotubes. If so, we would have expected a decrease in the expression of Cavin-1 and SOCS3 levels. New Supplementary Fig. 6a shows that Cavin-1 and SOCS3 levels are not decreased in mutant myotubes. On the contrary, SOCS3 total expression is significantly higher in P28L and R26Q mutant myotubes compared to WT myotubes. This result is consistent with the observed increase of SOCS3 gene expression in mutant myotubes (Fig. 3g). While additional experiments would be required to thoroughly address this point, these preliminary data suggest that the increased activation of STAT3 in mutant myotubes is unlikely to occur through the SOCS3-Cavin-1 interaction that has been uncovered in endothelial cells. Indeed, by using peptides mimicking the caveolin scaffolding domain (CSD), we show now that the regulation of STAT3 activation requires the CSD domain (Supplementary Fig. 6c, d).

We thank the reviewer for bringing to our attention these studies that are now discussed in the revised version of the manuscript (p.12).

• Will the paper be of interest to others in the field?

Mainly interesting for scientists working with these specific Caveolin-3 mutants. Although the principal subject is not totally new, the paper is of some interest to others in the field (cell biologists) as the function of caveolae is still under debate.

We agree with the reviewer that our findings will be of interest for scientists working with Cav3-related muscle diseases.

The reviewer rightly points out that although caveolae play a major role in signaling, this function remains debated. In response to the reviewer questions below, we believe that this regulation could be extended to Cav1 in non muscle cells and to other signaling pathways. The interest of our study goes therefore well beyond Cav3 muscle diseases and IL6/STAT3 signaling as it uncovers a new mechanism by which the reactivity of caveolae to mechanical stress is tightly coupled to the regulation of intracellular signaling.

We also believe that our study will be of interest to the field of mechanobiology as there are few examples of mechanosignaling events that are regulated independently from the actin-myosin machinery.

• Will the paper influence thinking in the field?

The impact of caveolae on membrane flexibility and signalling is interesting. However, it remains to be shown that the effect of Caveolin-3 on IL-6 signalling is specific and does not affect all other signalling pathways as well.

As the reviewer pointed out above, caveolae have been associated with several signaling pathways. It is therefore likely that the mechanism uncovered here for IL6/STAT3 signaling in muscle cells could be extended, to some point, to other signaling pathways. Our new results (Supplementary Fig. 6d, e) showing that the caveolin scaffolding domain (CSD) is required for the control of IL6/STAT3 by Cav3 implies that signaling effectors interacting with Cav1 or Cav3 via the CSD domain are likely to follow the same regulation. While testing “all other signaling pathways” in human myotubes is beyond the scope of this manuscript, we nevertheless investigated several additional signaling pathways. Figure R1 below shows that the EGF/MAPK signaling pathway was not affected by Cav3 in human myotubes. On the other hand, our

preliminary data show that in endothelial cells, the STAT3 signaling pathway follows the same regulation by Cav1 since STAT3 is hyperactivated at steady state in Cav1^{-/-} endothelial cells. These new data reveal a certain degree of selectivity that is now discussed in the revised manuscript (p.13-14).

In addition, a previous study performed in Cav3 p.P104L transgenic mice revealed significant changes at the proteomic level between Cav3 p.P104L transgenic mice and WT littermates (PMID: 30153853). This mutation results also in the sequestration of Cav3 in the Golgi apparatus. We can therefore hypothesize that the P28L and R26Q mutation will also affect muscle cell protein expression and thereby impact directly or indirectly other signaling pathways in muscle cells. This study is now mentioned in the discussion of the revised manuscript (p.14).

• Are the claims convincing? If not, what further evidence is needed?

Claims and interpretations are correct. Just a few minor (technical, editorial) points need to be addressed:

Numbering in figure 1 and in the legend to figure 1 does not correlate. (d should be e, c should be d etc.)

We thank the reviewer for pointing out these errors and apologize for the inconvenience. The figure legend has now been modified accordingly (p.24).

Figure 2d and f: WT cells and siCtl cells strongly differ in respect to % burst cells and time of bursting, despite the fact that both are control cells. Reliability of the assay?

We agree with the reviewer that the percentage of burst cells and time of bursting differs between WT myotubes and WT myotubes transfected with scramble siRNA (siCtl). Differentiated human myotubes are notoriously refractory to transfection. In order to achieve a full extinction of Cav3, we had to transfect these cells twice with a lipid-based reagent (HiPerfect, Qiagen). We

think that this treatment is likely to modify the mechanical properties of the plasma membrane thereby explaining that siCtl myotubes do not respond exactly as non transfected WT myotubes.

Regarding the reliability of our assay, our data show very low standard variation between replicates (even when repeated several months after the initial experiment) for each given condition (Figs. 2c, e and 6a; Supplementary Fig. 4b). Finally, to answer Reviewer #3, we performed new experiments with the bursting assay (see Fig. R4 below) and the new data show a mean percentage of burst for siCtl myotubes ($26\% \pm 2$) that is almost identical to the percentage reported in Fig. 2e ($23\% \pm 1$).

Figure 3a and 4a: STAT3 expression is strongly increased in Caveolin-3 P28L expressing cells. Please comment!

We also noticed the higher level of STAT3 expression in Cav3-P28L myotubes. Variations in STAT3 levels have been frequently observed under pathological contexts such as cancer². While variations in STAT3 levels play certainly a role in STAT3-dependent signaling, this is the phosphorylation form of STAT3 which is important for its nuclear translocation and gene regulation.

In the particular case of our study, we can rule out STAT3 increased levels as responsible for the IL6/STAT3 signaling defects observed in P28L myotubes since we found similar defects in R26Q mutant myotubes even though they present STAT3 levels identical to WT myotubes. We have now discussed this in the revised manuscript (p.7).

Figures 3a,b,e,f, Figure 4, S3: Obviously, STAT3 and (p)STAT3 staining was not from the same gel (no stripping of the membrane) but from two gels running in parallel. Thus, each gel has its own tubulin staining for quantification. Where is the 2nd tubulin staining? $(pSTAT3/tub) / (STAT3/tub) = (pSTAT3/STAT3)$ if only one tubulin staining has been considered!

In the case of two separate gels for (p)STAT and STAT calculation should be: $(pSTAT3/tub^{(pSTAT)}) / (STAT3/tub^{(STAT)})$ because the tubulin signal from the pSTAT-gel differs from the tubulin signal on the STAT-gel.

In case only one gel was used and been stripped after (p)STAT staining, $(pSTAT3/tub) / (STAT3/tub) = (pSTAT3/STAT3)$ and tubulin signals are meaningless.

We thank the reviewer for his/her helpful comments. We have now modified our manuscript accordingly. For all mentioned figures (now Figs. 3a, e, 4a, c and Supplementary Fig. 3a), we show the alpha-tubulin bands that were missing. Since STAT3 and pSTAT3 signals are obtained from two different gels, the phosphorylation status was calculated as follows: $(pSTAT3/tubulin^{pSTAT3}) / (STAT3/tubulin^{STAT3})$. This calculation is now detailed in the Methods section.

Figure S1 c: please indicate in the figure legend that the figures correspond to the identical figures shown in figure 1e.

The figure legend has been modified accordingly (p.24, p.31).

• Are there other experiments that would strengthen the paper further? How much would they improve it, and how difficult are they likely to be?

Studying the mode of inhibition of IL-6 signalling would be interesting, but not absolutely necessary. (receptor internalization, SOCS3-mediated inhibition, phosphatases ...?)

We agree with the reviewer that studying the mode of inhibition of IL-6 signaling while interesting would be beyond the scope of the current manuscript. We nevertheless explored some key

elements of the IL-6 receptor pathway (Supplementary Fig. 6). First, we investigated the total protein expression of IL6R α and gp130, the two subunits encompassing the IL6 receptor. Supplementary Fig. 6a shows that both subunits are expressed to similar levels in P28L, R26Q mutant and WT myotubes. These preliminary data suggest that the increased activation of STAT3 in mutant myotubes is not caused by a higher amount of IL6 receptors in these cells. In addition, we monitored IL6R complex localization within caveolae at the plasma membrane. In P28L and WT myotubes expressing Cav3-GFP, the Cav3-GFP fluorescent signal did not colocalize with immunofluorescence surface staining neither for gp130 (Supplementary Fig. 6, upper panel) nor for IL6R α (Supplementary Fig. 6, lower panel). In agreement with these data, we failed to immunoprecipitate gp130 with endogenous Cav3 (not shown). Whereas these results do not claim for a direct interaction between the IL6R complex with Cav3 in caveolae, we found that cavtratin, a small peptide mimicking the caveolin scaffolding domain (CSD) was able to significantly reduce the hyperactivation of the IL6/STAT3 pathway in Cav3 P28L mutant myotubes (Supplementary Fig. 6c, d). The CSD has been proposed to regulate several signaling molecules such as eNOS through a direct interaction (Byrne et al., PLoS One 2012, PMID:23028656). All together, these data suggest that the regulation of IL6 mechanosignaling may require the interaction with Cav3-CSD, probably outside of caveolae. Finally, we followed the reviewer suggestion to address a potential role of a SOCS3-mediated inhibition. Unfortunately, we were not able to silence properly SOCS3 expression by siRNA (using Qiagen FlexiTube GeneSolution GS9021, pool 4 siRNA).

• Are the claims appropriately discussed in the context of previous literature?

There are some more data available on osmo-sensing and IL-6 signalling, caveolae and IL-6 sensing, and caveolae and membrane tension. However, the most important studies have been considered.

While we agree with the reviewer that some studies had suggested a possible link between IL-6 signaling and osmolarity (see our response to the reviewer's first concern above), our study is the first one to show that the control of IL6/STAT3 signaling is tightly coupled to the mechanical response of caveolae.

• If the manuscript is unacceptable in its present form, does the study seem sufficiently promising that the authors should be encouraged to consider a resubmission in the future?

Only very minor points need to be corrected.

• Is the manuscript clearly written? If not, how could it be made more accessible?

The paper is very well written and easy to read. Presentation of data is very clear.

• Could the manuscript be shortened to aid communication of the most important findings?

No.

• Have the authors done themselves justice without overselling their claims?

No

• Have they been fair in their treatment of previous literature?

See above

We have included and discussed the new references mentioned by the reviewer in the revised version.

• Have they provided sufficient methodological detail that the experiments could be reproduced?

Yes, but some figure legend could be improved. Some stainings are missing (see above)

• *Is the statistical analysis of the data sound?*

Yes (see above)

• *Should the authors be asked to provide further data or methodological information to help others replicate their work? (Such data might include source code for modelling studies, detailed protocols or mathematical derivations).*

No

• *Are there any special ethical concerns arising from the use of animals or human subjects?*

No concerns in this respect.

Reviewer #2

In this manuscript, the authors examine the effect of caveolae on the response of myotubes to mechanical stress. They study myotubes expressing either of two non-functional caveolin-3 mutants, associated with different muscle diseases, and also suppress caveolin-3 expression in wild-type myotubes with siRNA.

The technical quality of the experiments is very good, and (with the caveat detailed below), the results are believable. The main issue for publication lies in the significance of the findings.

We are grateful to the reviewer for his/her positive comments on the quality of our results. In the revised version, we have endeavored to strengthen the significance of our data.

The authors have already published the most important results shown in Figs. 1 and 2. Specifically, in their seminal 2011 Cell paper (the first report that caveolae protect muscle cells against mechanical stress), the authors showed that myotubes expressing Cav3 P28L, which is retained in the Golgi, fail to buffer the rise in membrane tension in response to mechanical stress, and that they rupture more easily than normal myotubes during hypo-osmotic shock. The new findings shown in Figs. 1 and 2 are that cells expressing Cav3-P28L and Cav3-R26Q have reduced levels of cell-surface caveolae, and that Cav3-R26Q (also retained in the Golgi) has the same effects as Cav3-P28L. These are expected results.

We agree with the reviewer that these results were expected. We believe however that it was important to show that the Cav3-R26Q mutant presented the same caveolae-related membrane protection defects than the Cav3-P28L mutant.

Figure 5 shows that the defects in Cav3-P28L myotubes can be overcome by expression of WT Cav3-GFP. This is an important control, but in itself does not contribute important new information.

Indeed, this experiment was initially meant as a control. The importance of this experiment is further illustrated by our new analysis of this condition by immunofluorescence (Fig. 6e, f) where we show that Cav3 re-expression in mutant myotubes restores the mechanoregulation of IL6/STAT3 signaling by caveolae. Furthermore, our *in vivo* experiment with myoblasts transplantation in mouse muscles (Supplementary Fig. 5, described in response to reviewer #3, question 3, p.12) confirmed the rescue results obtained *in vitro* (Fig. 5b, c) since Cav3-GFP was located at the plasma membrane after fusion of Cav3-GFP-rescued P28L human myoblasts with mouse fibers.

The most significant results are shown in Figures 3 and 4. Myotubes normally respond to IL6 by increasing phosphorylation of STAT3. The authors show that myotubes expressing the Cav3 mutants have constitutively higher levels of pSTAT3 than WT, and that the IL6-induced increase in pSTAT3 levels is further enhanced in the mutants, and also in siCav3 myotubes. Four transcriptional targets of pSTAT3 are also constitutively up-regulated in the mutants. This is the first demonstration that IL6 signaling is negatively regulated by caveolae in muscle cells. Given the important role of IL6 signaling in muscle physiology and disease, this is a novel and significant result, and probably justifies publication.

We thank the reviewer for his/her positive appreciation of the novelty and significance of the main finding of our study.

This finding is extended in Figure 4 and Supplementary Figure 3. The authors first present evidence that prior mechanical stress (hypo-osmotic shock or stretch) inhibits subsequent STAT3 phosphorylation in response to IL6 treatment in normal myotubes. Next, they show that this inhibitory effect is lost in myotubes lacking caveolae (Cav3 mutants or Cav3 siRNA). While interesting, the significance of these findings is tempered a bit by the quality of the data. I am not aware of any prior demonstration that mechanical stress normally inhibits the IL6 response in muscle. Figures 4a and 4b show pSTAT3 levels in WT myotubes after 0 or 5 min of IL6 treatment, with or without prior hypo-osmotic shock. The authors report a robust 80% inhibition in IL6-mediated stimulation of STAT3 phosphorylation by hypo-osmotic shock (though all the relevant bands on the representative blot shown are quite faint).

However, in Figure 4c, they repeat the same experiment in siRNA-treated cells. Myotubes treated with control siRNA (expected to be the same as the control cells in Figure 4a) show a much less impressive (though still significant to $p < 0.05$) inhibition of IL6-induced STAT3 phosphorylation by prior hypo-osmotic shock than the control myotubes in Figure 4a. Together with the overall unevenness of the bands in the blots shown, this discrepancy somewhat reduces the impact of the finding.

We agree that the decrease of IL6-induced STAT3 phosphorylation under hypo-osmotic shock is lower in siCtl myotubes than in WT myotubes. This is in agreement with the apparent higher resistance of these cells to mechanical constraints (see Fig. 2).

To rule out the possible interference of undifferentiated myotubes, which express Cav1 and not Cav3, we directly monitored by immunofluorescence the nuclear translocation of pSTAT3 in multinucleated differentiated myotubes in response to IL6 stimulation. These new data (panels e and f of Fig. 4) show that after hypo-osmotic shock, differentiated siCtl myotubes exhibit a significant decrease ($p < 0,01$) of IL6-induced pSTAT3 nuclear translocation compared to iso-osmotic conditions.

We also performed new immunofluorescence experiments (Fig. 6e, f) showing that overexpression of Cav3-GFP in Cav3-P28L mutant myotubes leads to a significant decrease of IL6-induced pSTAT3 nuclear translocation under hypo-osmotic conditions in contrast to Cav3-P28L myotubes expressing GFP only. These new data confirm that the re-expression of Cav3 in mutant myotubes can restore the mechanoregulation of IL6/STAT3 signaling by caveolae.

Minor points.

1. The authors should fix the legend to Figure 1. They apparently merged the original panels c and d, and eliminated the original panel e, without changing the legend.

We thank the reviewer for pointing out this error and apologize for the inconvenience. The figure legend has now been modified accordingly (p.24).

2. The authors should comment on why mutant caveolin-3 retains endogenous caveolin-1 in the Golgi (Supp. Fig. 1), but WT-Cav3-GFP overexpressed in mutant cells is transported to the cell surface (Figure 5).

We addressed the reviewer's concern by performing new experiments where we monitored the behavior of Cav1 relatively to the level of Cav3 expression (Fig. R2). By immunoprecipitation of endogenous Cav3, we observed that both WT and mutant Cav3 interact to the same extent with endogenous Cav1 (Fig. R2a). These data support our observation that Cav1 is also retained in the Golgi apparatus of P28L and R26Q mutant myotubes (Supplementary Fig. 1c). As shown in Fig. R2b, Cav3-GFP is expressed to a much higher amount than endogenous Cav3 in transduced WT and P28L mutant myotubes. We believe therefore that overexpressed Cav3-GFP is able to be transported to the plasma membrane of P28L mutant myotubes by overwhelming the amount of endogenous mutated Cav3 retained in the Golgi apparatus.

Figure R2 | Relative amounts of Cav3 and Cav1 levels in WT and P28L myotubes before and after Cav3-GFP transfection.

(a) Cell lysates of WT, P28L or R26Q myotubes were incubated with anti-GFP (negative control) or anti-Cav3 antibody to immunoprecipitate (IP) endogenous Cav3 and reveal co-immunoprecipitated endogenous Cav1. (b) Immunoblot analysis of total levels of Cav3, Cav3-GFP and Cav1 in WT or P28L mutant myotubes overexpressing GFP or Cav3-GFP. Clathrin heavy chain (CHC) serves as a loading control. Reproducibility of experiments: (a) Representative data of 2 independent experiments. (c) Show representative data of 3 experiments.

3. Figure 2c and d. They should say how the mean time of bursting was calculated.

The calculation of the bursting time is now described in the Methods section.

4. Total STAT3 levels are much higher in Cav3-P28L myotubes than in WT or Cav3-R26Q myotubes. The authors might comment on this.

We also noticed the higher level of STAT3 expression in Cav3-P28L myotubes. Variations in STAT3 levels have been frequently observed under pathological contexts such as cancer². While variations in STAT3 levels play certainly a role in STAT3-dependent signaling, this is the

phosphorylation form of STAT3 which is important for its nuclear translocation and gene regulation.

In the particular case of our study, we can rule out STAT3 increased levels as responsible for the IL6/STAT3 signaling defects observed in P28L myotubes since we observe similar defects in R26Q mutant myotubes although they present STAT3 levels identical to WT myotubes. This is now discussed in the revised manuscript (p.7).

5. Quantifications of bands in blots are expressed as $(pSTAT3/tubulin)/(STAT3/tubulin)$. This is mathematically the same as $pSTAT3/STAT3$, which is simpler and avoids the possible complication of slight spurious variations in tubulin signal between lanes.

We thank the reviewer for his/her helpful comments. We have now modified our manuscript accordingly. For all mentioned figures (now Figs. 3a, e, 4a, c, and Supplementary Fig. 3a), we show the alpha-tubulin bands that were missing. Since STAT3 and pSTAT3 signals are obtained from two different gels, the phosphorylation status was calculated as follows: $(pSTAT3/tubulin^{pSTAT3})/(STAT3/tubulin^{STAT3})$. This calculation is now detailed in the Methods section.

6. Fig. 3c and d. Why did the authors quantitate the pSTAT3 nucleus:cytoplasm ratio, instead of the nucleus:cytoplasm ratio of total STAT3? The regulated step in activation is STAT3 phosphorylation, not nuclear import of pSTAT3.

Many laboratories including ours commonly use pSTAT nuclear translocation to monitor the level of JAK/STAT activation by different cytokines^{3,4}. Furthermore, previous studies^{5,6}, and review⁷) have shown that non phosphorylated STAT3 can shuttle between the cytosol and the nucleus with no effect on gene transcription. The nucleus:cytoplasm ratio of total STAT3 is therefore less accurate to monitor JAK/STAT activation by IL6.

Nevertheless, to address the reviewer concern, we compared the immunofluorescence staining obtained for STAT3 and pSTAT3 in human myotubes (Fig. R3).

Figure R3 | STAT3 and pSTAT3 immunofluorescence staining in IL6 stimulated myotubes

Immunofluorescent labeling of STAT3 and pSTAT3 in WT, Cav3 P28L or Cav3 R26Q myotubes stimulated by IL6 for 15 min and analyzed by confocal microscopy. Scale bar = 10 μ m. Data are representative of two experiments.

In our hands, after IL6 stimulation, we detect little difference between cytosolic and nuclear STAT3. In contrast, pSTAT3 is more clearly detected in myotube nuclei. This difference may be due to differences between STAT3 and pSTAT3 antibodies and this is why we believe that following pSTAT3 nuclear translocation is more accurate to assess IL6-induced JAK/STAT activation.

Is the OT value for WT meaningful, given the low levels of pSTAT3 (“little tyrosine phosphorylation of STAT3, if any”)? They should also state explicitly that the blue stain in Fig. 3c is DAPI.

We believe that the OT value is meaningful as it allows to compare the level of pSTAT3 nuclear translocation between mutated (P28L and R26Q) and WT myotubes at steady state. In Fig. 3c, we have now modified image labeling to state that the blue color corresponds to DAPI.

7. Fig. 3e and f. They should re-write text to tone down the conclusions a bit to accurately reflect the data. There's no significant constitutive activation of STAT3 without IL6 addition in Cav3 siRNA-treated cells. For this reason, the depletion of cav1 in WT myotubes doesn't exactly reproduce the phenotype of the Cav3 mutants. These results do NOT "demonstrate that the absence of Cav3 and/or caveolae at the PM of mutant myotubes is responsible for the constitutive hyperactivation of the IL6/STAT3 signaling pathway".

We agree with the reviewer that Cav3 siRNA-treated myotubes do not present a significant constitutive activation of STAT3 as observed in P28L and R26Q mutant myotubes. Since myotubes express both Cav1 and Cav3, we hypothesized that Cav1 could compensate for the absence of Cav3 at the plasma membrane of siRNA-Cav3 myotubes. Indeed, Cav1 was mainly localized in the Golgi apparatus and barely detectable at the plasma membrane of mutant myotubes by immunofluorescence (Supplementary Fig. 1c). We performed new experiments by isolating giant plasma membrane vesicles (GPMVs), so as to restrict the detection of Cav3 and Cav1 at the plasma membrane (Supplementary Fig. 1d, e). The analysis by western blot of GPMV-enriched fractions, which were devoid of the Golgi apparatus (GM130) and nucleus (lamin A/C) protein markers, confirmed that Cav3 and Cav1 were present at the plasma membrane in P28L and R26Q myotubes to a much lesser extent than in WT myotubes.

On the contrary, Cav1 was still localized at the plasma membrane of Cav3 depleted WT myotubes (Fig. R4). It is therefore likely that the presence of Cav1 at the plasma membrane of siCav3 WT myotubes may contribute to limit the STAT3 hyperactivation by a mechanism similar to Cav3. We have now re-written this part of our manuscript (p. 7).

Figure R4 | Cav1 expression and localization in Cav3 depleted WT myotubes.

(a) Immunoblot analysis of total levels of Cav3 and Cav1 in WT ctl (siCtl) and Cav3-depleted (siCav3) myotubes. (b) Immunofluorescent labeling of Cav3, Cav1 and the Golgi marker GM130 in WT ctl (siCtl) and Cav3-depleted (siCav3) myotubes analyzed by confocal microscopy. Arrows indicate the plasma membrane. (b) Scale bar = 10 μ m. Reproducibility of experiments: (a, b) Representative data of 2 independent experiments.

8. Discussion. Line 12: should be sensitivity, not sensibility. Line 15; should be restore, not reinstall.

We thank the reviewer for noting these errors that have now been corrected in the revised manuscript (p.11).

Reviewer #3:

The present work by Dewulf et al investigates the role of Caveolin-3 in mechanosensing of human skeletal muscle cells and regulation of IL-6/STAT3 signaling. The authors show that myotubes derived from patients with P28L and R26Q mutations in the Caveolin-3 gene, which result in multiple human disease manifestations, including hyperCKemia, rippling muscle disease and neuropathies, exhibit a reduction in the density of caveolae on the surface of the cells. Mutant proteins are mainly retained in the Golgi instead of cell surface localization. Exposure of mutant myotubes to hypo-osmotic shock shows that mutant cells are impaired in their ability to cope with mechanical stress. The authors further link absence of Cav-3 and reduction of caveolae to a dysregulation of IL-6/STAT3 signaling. Mutations in caveolin-3 are associated with a broad spectrum of human diseases, thus understanding the mechanism underlying the effects of these mutations on diseases is relevant. While it has been previously shown that mutant Cav-3 proteins accumulate in the Golgi and lead to a reduced number of caveolae, the novelty of the current work is the link with mechanosensing and the effects on IL-6/STAT3 signaling in human muscle cells. The manuscript is well written.

We thank the reviewer for her/his positive comments.

However additional experiments are required in order to fully support the authors' interpretation. Specifically, more mechanistic depth on the link between Cav-3, caveolae and IL-6/STAT3 signaling as well as the potential impact of STAT3 on the observed phenotype should be provided. In addition, it would be useful to provide validation of the in vitro findings in vivo by cell transplantation assays into mice.

We have endeavored to provide more compelling data in the revised version of our work.

Major points:

1- Mechanism connecting Cav-3 and caveolae to IL-6/STAT3 signaling: The authors cite previous work in which the gp130 receptor has been shown to localize in caveolae in a myeloma cell line. This should be shown also in differentiated human myotubes. In addition, it is unclear why the absence of Cav-3 or caveolae leads to hyperactivation of the IL-6 STAT3 signaling. Are the IL-6 receptor levels on the surface of the cells increased in the mutants vs wild-type cells? Or is it due to intracellular signaling? Or to direct interaction Cav-3/STAT3 or Cav-3/gp130? Increasing the depth of the molecular mechanism underlying the defect in the IL-6/STAT3 signaling in these human mutant cells would significantly strengthen this work.

We provide now new experimental data regarding the molecular mechanisms that could be involved in the mechanoregulation of IL6/ STAT3 signaling by caveolae (see Supplementary Fig. 6). First, we investigated the total protein expression of IL6R α and gp130, the two subunits encompassing the IL6 receptor. Supplementary Fig. 6a shows that both subunits are expressed to similar levels in P28L, R26Q mutant and WT myotubes. These preliminary data suggest that the increased activation of STAT3 in mutant myotubes is not caused by a higher amount of IL6 receptors in these cells. In addition, we monitored IL6R complex localization within caveolae at

the plasma membrane. In P28L and WT myotubes expressing Cav3-GFP, the Cav3-GFP fluorescent signal did not colocalize with immunofluorescence surface staining neither for gp130 (Supplementary Fig. 6, upper panel) nor for IL6R α (Supplementary Fig. 6, lower panel). In agreement with these data, we failed to immunoprecipitate gp130 with endogenous Cav3 (not shown). Whereas these results do not claim for a direct interaction between the IL6R complex with Cav3 in caveolae, we found that cavtratin, a small peptide mimicking the caveolin scaffolding domain (CSD) was able to significantly reduce the hyperactivation of the IL6/STAT3 pathway in Cav3 P28L mutant myotubes (Supplementary Fig. 6c, d). The CSD has been proposed to regulate several signaling molecules such as eNOS through a direct interaction (Byrne et al., PLoS One 2012, **PMID:23028656**). All together, these data suggest that the regulation of IL6 mechanosignaling may require the interaction with Cav3-CSD, probably outside of caveolae. This is now discussed in the revised manuscript (p.13, 14).

Finally, we followed the reviewer suggestion to address a potential role of a SOCS3-mediated inhibition. Unfortunately, we were not able to silence properly SOCS3 expression by siRNA (using Qiagen FlexiTube GeneSolution GS9021, pool 4 siRNA).

2- Impact of STAT3 signaling on the observed phenotype: this is an underdeveloped aspect of the current work: is STAT3 signaling involved in the defect in mechanosensing and consequent bursting or is it unrelated?

We followed the reviewer suggestion to address the role of STAT3 signaling in mechanoprotection. We depleted WT myotubes from STAT3 by siRNA and challenged them by hypo-osmotic shock (Fig. R5). Unexpectedly, we found that STAT3 depletion led to a higher percentage of burst cells and mean bursting time compared to siCtl myotubes. To further analyze the increased membrane fragility observed in siSTAT3 myotubes, we measured Cav1 and Cav3 protein expression levels. We found that the levels of Cav3 expression were significantly decreased upon STAT3 depletion, which is likely to explain, at least partially, the increased membrane fragility, as it phenocopies the knock-down of Cav3. Interestingly, Cav1 levels remain unchanged.

While these results are consistent with the data present in our study, we believe that they should be pursued in follow-up studies.

Figure R5 | STAT3 and membrane integrity.

(a) Micropatterned WT ctl (siCtl) and STAT3-depleted (siSTAT3) myotubes were loaded with calcein-AM (green). The medium was switched to a 30 mOsm medium supplemented with propidium iodide (PI, red). Representative pictures were taken at the indicated times during hypo-osmotic shock. Arrows correspond to myotubes and asterisks correspond to burst myotubes. **(b)** Quantification of the percentage of burst myotubes (upper panel) and mean time of bursting in minutes (lower panel) in **(a)**. **(c)** Immunoblot analysis and quantification of total levels of STAT3, Cav3 and Cav1 in WT ctl (siCtl) and STAT3-depleted (siSTAT3) myotubes. Tubulin serves as a loading control. **(a)** Scale bar = 120 μ m. Reproducibility of experiments: **(a)** Representative data of 3 independent experiments quantified in **(b)** (% burst cells: siCtl n = 334, siSTAT3 n = 216; mean time of bursting: siCtl n = 89, siSTAT3 n = 157). **(c)** Show representative data of 3 experiments. Mean value \pm SD. **(b, c)** Statistical analysis with two-tailed unpaired t test; * P<0,05; *** P<0,001, ns non significant.

3- All the presented work has been performed *in vitro*, while it would be useful to also test in transplantation assays whether mutant and healthy human muscle cells behave differently when injected intramuscularly into recipient mice.

We followed the reviewer suggestion by performing *in vivo* experiments in mice taking advantage of the muscle expertise of our collaborators (Anne Bigot, Mona Bensalah, Elisa Negroni who are now new authors on the revised manuscript) at the Institute of Myology in Paris. We transplanted WT GFP, P28L GFP or P28L Cav3-GFP transduced human myoblasts in *tibialis anterior* (TA) mouse muscles previously exposed to cryodamage to induce severe muscle damage. Three weeks after transplantation, the recipient muscles were dissected and immunostained for specific human and mouse protein markers (human spectrin and lamin A/C; human and mouse laminin) (New Supplementary Fig. 5). For the three types of transplanted cells, we observed mature muscle fibers containing human spectrin protein and human lamin A/C positive nuclei, indicating the proper fusion of human myoblasts with regenerating mouse myofibers. No significant difference was observed regarding the number of positive cells for human lamin A/C or the number of positive fibers for human spectrin.

These results indicate that the immortalized Cav3-P28L mutated myoblasts have maintained their potential to differentiate *in vivo* after transplantation into regenerating muscles of mice. The capacity to maintain their differentiation program was also documented for several muscular

dystrophies including Duchenne and facioscapulohumeral muscular dystrophies, even though they present more severe clinical disorders than the Cav3-related muscle dystrophies studied here (**PMID: 22040608**).

These experiments are nevertheless informative since the Cav3-P28L mutant human myotubes had not been characterized with this transplantation assay before. Furthermore, our *in vivo* data revealed that after fusion of Cav3-GFP-rescued P28L myoblasts with mouse fibers, Cav3-GFP is located at the plasma membrane. This result demonstrates that the proper localization of Cav3 can be rescued *in vivo*, which may open further investigations for therapeutic applications.

4- In Figure 3e-f the transient knockdown of Cav3 leads to hyperactivation of STAT3 upon IL-6 stimulation, but not at steady state, while the mutant cells exhibit elevated STAT3 signaling also at steady state. Can the authors comment on this different phenotype?

We agree with the reviewer that Cav3 siRNA-treated myotubes do not show hyperactivation at steady state in contrast to P28L and R26Q mutant myotubes. Since myotubes express both Cav1 and Cav3, we hypothesized that Cav1 could compensate for the absence of Cav3 at the plasma membrane of siRNA-Cav3 myotubes. Indeed, Cav1 was mainly localized in the Golgi apparatus and barely detectable at the plasma membrane of mutant myotubes by immunofluorescence (Supplementary Fig. 1c). We performed new experiments by isolating giant plasma membrane vesicles (GPMVs), so as to restrict the detection of Cav3 and Cav1 at the plasma membrane (Supplementary Fig. 1d, e). The analysis by western blot of GPMV-enriched fractions, which were devoid of the Golgi apparatus (GM130) and nucleus (lamin A/C) protein markers, confirmed that Cav3 and Cav1 were present at the plasma membrane in P28L and R26Q myotubes to a much lesser extent than in WT myotubes.

On the contrary, Cav1 was still localized at the plasma membrane of Cav3 depleted WT myotubes (Figure R4). It is therefore likely that the presence of Cav1 at the plasma membrane of siCav3 WT myotubes may contribute to limit the STAT3 hyperactivation by a mechanism similar to Cav3. We have now re-written this part of our manuscript (p. 7).

5- The two Cav-3 mutations assessed lead to different disease manifestations, thus it would be useful to include a discussion of how the observed phenotype reported here might contribute to the two different conditions.

While it would have been tempting to link our findings to the observed phenotypes in the patients bearing P28L and R26Q mutations, we think it would be very speculative. Actually, the R26Q mutation has been associated with four different muscular dystrophies, namely HyperCKemia (HCK), rippling muscle disease (RMD), distal myopathies (DM), hyperCKemia (HCK) and limb-girdle muscular dystrophy 1C (LGMD-1C). The P28L mutation has been described only for HCK. In our hands, the human myotubes from patients with P28L and R26Q Cav3 mutations show the same loss of caveolae mechanoprotection and mechanoregulation of IL6/STAT3 signaling by Cav3. Thus, linking our conclusions to the different disease manifestations could be seen as overinterpreted.

Specific Points:

1- In Figure 1d and Suppl. Figure 1c, a quantification of the colocalization between Cav-3 and GM130 and Cav-1 in the human cells should be shown, in order to substantiate the authors' interpretation, or the findings remain only qualitative.

We have now added the Manders' colocalization coefficient for immunostaining of the Golgi marker GM130 with either Cav3 or Cav1 for each myotube cell line in Figure 1d (Cav3/GM130) and Supplementary Fig. 1d (Cav1/GM130). It further confirms our findings with P28L and R26Q

Cav3 are mainly localized in the Golgi apparatus together with Cav1 whereas Cav3 and Cav1 are both localized at the plasma membrane in WT myotubes.

2- In Figure 3b statistical significance of the difference in STAT3 phosphorylation should be indicated in both figure and figure legend.

Unfortunately, due to the high variability of the signals obtained by WB, we could not obtain a statistical significance despite a clear tendency of increased pSTAT3/STAT3 levels in mutant myotubes. To rule out the possible interference of undifferentiated myotubes, which express Cav1 and not Cav3, we directly monitored by immunofluorescence the nuclear translocation of pSTAT3 in multinucleated differentiated myotubes in response to IL6 stimulation (Fig. 3c, d). These data show that P28L and R26Q myotubes exhibit a significant increase ($p < 0,05$) of IL6-induced pSTAT3 nuclear translocation compared to WT myotubes.

Additional references

1. Bode JG, Gatsios P, Ludwig S, Rapp UR, Häussinger D, Heinrich PC, Graeve L. The mitogen-activated protein (MAP) kinase p38 and its upstream activator MAP kinase kinase 6 are involved in the activation of signal transducer and activator of transcription by hyperosmolarity. *J Biol Chem*, **274**:30222-30227 (1999). **PMID: 10514514**
2. Banerjee, K., and H. Resat. Constitutive activation of STAT3 in breast cancer cells: A review. *Int J Cancer*. **138**:2570-2578 (2016). **PMID: 26559373**
3. Marchetti M, Monier MN, Fradagrada A, Mitchell K, Baychelier F, Eid P, Johannes L, Lamaze C. Stat-mediated signaling induced by type I and type II interferons (IFNs) is differentially controlled through lipid microdomain association and clathrin-dependent endocytosis of IFN receptors. *Mol Biol Cell*, **17**:2896-2909 (2006). **PMID: 16624862**
4. Blouin CM, Hamon Y, Gonnord P, Boullaran C, Kagan J, Viaris de Lesegno C, Ruez R, Mailfert S, Bertaux N, Loew D, Wunder C, Johannes L, Vogt G, Contreras FX, Marguet D, Casanova JL, Galès C, He HT, Lamaze C. Glycosylation-Dependent IFN- γ R Partitioning in Lipid and Actin Nanodomains Is Critical for JAK Activation. *Cell*, **166**:920-934 (2016). **PMID: 27499022**
5. Liu L, McBride KM, Reich NC. STAT3 nuclear import is independent of tyrosine phosphorylation and mediated by importin- α 3. *Proc Natl Acad Sci U S A*. **102**:8150-8155 (2005). **PMID: 15919823**
6. Vogt M, Domszalai T, Kleshchanok D, Lehmann S, Schmitt A, Poli V, Richter W, Müller-Newen G. The role of the N-terminal domain in dimerization and nucleocytoplasmic shuttling of latent STAT3. *J Cell Sci*. **124**:900-90 (2011). **PMID: 21325026**
7. Stark GR, Cheon H, Wang Y. Responses to Cytokines and Interferons that Depend upon JAKs and STATs. *Cold Spring Harb Perspect Biol*. **10** (2018). pii: a028555. **PMID: 28620095**

REVIEWERS' COMMENTS:

Reviewer #1 (Remarks to the Author):

The authors significantly improved their manuscript.
The new information and experimental data are very well appreciated.

All points of my first review have been addressed satisfactorily.

I have no additional concerns in respect to this really nice study.

Reviewer #2 (Remarks to the Author):

The authors have been extremely careful and meticulous in addressing my concerns, which were fairly minor. This revised manuscript is acceptable for publication.

Reviewer #3 (Remarks to the Author):

The authors have done a great job in addressing the reviewers' concerns, and as a result the resubmitted manuscript is significantly improved.

Remaining minor points:

1- Zoomed images in Figures 1d and Supplementary 6b are pixelated (low resolution). Please provide higher resolution immunofluorescence images.

2- Supplementary Fig. 4b is not cited or described in the text, and it appears to show that overexpression of WT Cav3-GFP increases the percentage of burst cells, which is inconsistent with the rest of the manuscript. Maybe it is a mislabeling issue in the Figure.

Response to reviewers (2):

We would like to thank the reviewers for their positive comments and for having helped us to improve our manuscript by their remarks and questions. The reviewers' comments appear in red, our point-to-point responses are in black. All the figure/table numbers are those of the revised manuscript.

Reviewer #1

The authors significantly improved their manuscript.
The new information and experimental data are very well appreciated.
All points of my first review have been addressed satisfactorily.
I have no additional concerns in respect to this really nice study.

We thank the reviewer for his/her positive appreciation of the novelty and significance of our study.

Reviewer #2

The authors have been extremely careful and meticulous in addressing my concerns, which were fairly minor. This revised manuscript is acceptable for publication.

We thank the reviewer for his/her positive comments.

Reviewer #3

The authors have done a great job in addressing the reviewers' concerns, and as a result the resubmitted manuscript is significantly improved.

We are grateful to the reviewer for his/her positive comments on the quality of our results.

Remaining minor points:

1- Zoomed images in Figures 1d and Supplementary 6b are pixelated (low resolution). Please provide higher resolution immunofluorescence images.

The microscopy images of Figure 1d are now provided with higher resolution in an .ai file through the submission website. We have also improved the quality of the images in Supplementary Figure 6b which were too compressed by the pdf conversion process.

2- Supplementary Fig. 4b is not cited or described in the text, and it appears to show that overexpression of WT Cav3-GFP increases the percentage of burst cells, which is inconsistent with the rest of the manuscript. Maybe it is a mislabeling issue in the Figure.

There was indeed a mistake in Supplementary Figure 4b. We have now corrected the scale of the graph to be consistent with the scale of the other bursting graphs (i.e. y axis up to 100%). The figure is now corrected in our manuscript and shows clearly that WT myotubes expressing Cav3-GFP have a bursting fraction (25%) identical to control cells i.e. WT myotubes transfected with siCtrl (see upper panel Fig 2f). Consistently, the bursting time of WT myotubes +GFP and

WT+Cav3-GFP myotubes are similar (Supp fig 4b). There is therefore no inconsistency with the rest of our manuscript.

In fact, there is slightly lower percentage of bursting in GFP-cells, which we believe is due to their higher level of cell differentiation (mean number of nuclei/cell: GFP = $6,45 \pm 0,25$; Cav3-GFP = $4,1 \pm 0,12$). This point has now been discussed in the revised version of our manuscript (p.10).